# Inhibition of host PARP1 contributes to the anti-inflammatory and antitubercular activity of pyrazinamide

Stefanie Krug [1,2], Manish Gupta[1,2], Pankaj Kumar[1,2], Laine Feller[2], Elizabeth A. Ihms[1,2], Bong Gu Kang [3,4], Geetha Srikrishna[1,2], Ted M. Dawson[3,4,5,6], Valina L. Dawson [3,4,5,7] & William R. Bishai[1,2] ✉

The antibiotic pyrazinamide (PZA) is a cornerstone of tuberculosis (TB) therapy that shortens treatment durations by several months despite being only weakly bactericidal. Intriguingly, PZA is also an anti-inflammatory molecule shown to specifically reduce inflammatory cytokine signaling and lesion activity in TB patients. However, the target and clinical importance of PZA's host-directed activity during TB therapy remain unclear. Here, we identify the host enzyme Poly(ADP-ribose) Polymerase 1 (PARP1), a pro-inflammatory master regulator strongly activated in TB, as a functionally relevant host target of PZA. We show that PZA inhibits PARP1 enzymatic activity in macrophages and in mice where it reverses TB-induced PARP1 activity in lungs to uninfected levels. Utilizing a PZA-resistant mutant, we demonstrate that PZA's immune-modulatory effects are PARP1-dependent but independent of its bactericidal activity. Importantly, PZA's bactericidal efficacy is impaired in PARP1-deficient mice, suggesting that immune modulation may be an integral component of PZA's antitubercular activity. In addition, adjunctive PARP1 inhibition dramatically reduces inflammation and lesion size in mice and may be a means to reduce lung damage and shorten TB treatment duration. Together, these findings provide insight into PZA's mechanism of action and the therapeutic potential of PARP1 inhibition in the treatment of TB.

*M. tuberculosis* (*M.tb*) is a highly successful human pathogen able to establish lifelong infections in spite of powerful inflammatory immune responses[1,2]. *M.tb* is first encountered by airway-resident myeloid cells which recruit and activate immune cells by initiating transcription factor signaling, notably NF-κB, to produce critical chemo- and cytokines, including TNFα, IL-12, and IL-1β[3,4]. TNFα and IL-12 then stimulate the antimicrobial activity of macrophages, CD4⁺ Th1 cell

differentiation, and production of the type 2 interferon IFNγ[4]. With the accumulation of *M.tb* antigen-specific CD4⁺ T cells, the host effectively restricts *M.tb* proliferation but is unable to clear the infection and resorts to surrounding infected foci with granulomatous structures to prevent further dissemination[4,5]. However, granulomas are associated with TB transmission, lung damage, poor drug penetration, and high antibiotic tolerance that make persisting bacilli difficult to kill[5–8].

[1]Center for Tuberculosis Research, Johns Hopkins University School of Medicine, Baltimore, MD, USA. [2]Department of Medicine, Johns Hopkins University School of Medicine, Baltimore, MD, USA. [3]Neuroregeneration and Stem Cell Programs, Institute for Cell Engineering, Johns Hopkins University School of Medicine, Baltimore, MD, USA. [4]Department of Neurology, Johns Hopkins University School of Medicine, Baltimore, MD, USA. [5]Solomon H. Snyder Department of Neuroscience, Johns Hopkins University School of Medicine, Baltimore, MD, USA. [6]Department of Pharmacology and Molecular Sciences, Johns Hopkins University School of Medicine, Baltimore, MD, USA. [7]Department of Physiology, Johns Hopkins University School of Medicine, Baltimore, MD, USA. ✉e-mail: wbishai1@jhmi.edu

Inflammatory TB immune responses are thus a double-edged sword that contains but fails to eliminate the infection, resulting in tissue damage and impaired treatment responses.

Pyrazinamide (PZA) has been a cornerstone of TB therapy for nearly 70 years but its mechanism of action remains incompletely understood[9–12]. Standard TB therapy consists of two months of PZA, ethambutol, isoniazid (INH), and rifampin (RIF), followed by an additional four months of INH and RIF. This combination of bactericidal (INH, ethambutol, RIF) and sterilizing (PZA, RIF) TB antibiotics is key for preventing drug resistance and treatment failures[13–15]. The sterilizing activity of a drug, defined as the speed with which the last remaining viable bacteria are killed, is assessed as the drug's contribution to the duration of treatment after which an infected host is considered free of viable bacteria and protected from relapse[14]. Even though PZA is administered only for two months and has minimal bactericidal activity compared to other TB antibiotics, it is an irreplaceable sterilizing component of standard TB therapy that reduces relapse rates and shortens treatment durations by several months[15–19]. While multiple models have been proposed to explain this perplexing discrepancy, none fully explain the unique treatment-shortening abilities of PZA[12]. A better understanding of PZA's key sterilizing activity could lead to the development of urgently needed improved TB treatment options.

PZA was developed as a TB antibiotic based on its structural similarity to vitamin B3 or nicotinamide (NAM), a compound with potent anti-inflammatory properties and moderate antitubercular activity in mice[9,10,20]. The discovery that PZA also modulates TB immune responses sparked the idea that PZA's sterilizing ability may derive from its unique combination of antitubercular and host-directed activities[21,22]. In M.tb-infected cells, mice, and patients, PZA dampens proinflammatory immune responses, including IFNγ, IL-1β and TNFα production, in a manner that suggests downregulation of NF-κB transcriptional activity but no host target of PZA has so far been identified[21,22]. Even though TNFα is essential for M.tb control, adjunctive TNFα inhibition during TB therapy can reduce lung pathology and treatment duration but most agents are costly and not orally bioavailable[23,24]. Identifying PZA's host target thus may reveal more applicable therapeutic approaches for TB.

While the clinical relevance of PZA's host effects remains controversial, several lines of evidence support the hypothesis that they might contribute to PZA's antitubercular activity. First, PZA is more bactericidal in M.tb-infected animals than in cells where it ranges from inactive to bacteriostatic[25–29]. Second, PZA has minimal efficacy in athymic nude mice, indicating that a functional immune system may be required for the antitubercular activity of PZA[30]. Third, PZA can restrict the growth of PZA-resistant M. bovis-BCG in macrophages but not in culture, suggesting that PZA may enhance host mechanisms of bacterial clearance[31]. Fourth, PZA had negligible bactericidal activity yet uniquely improved highly inflamed lung lesions in 14-day early bactericidal activity studies in TB patients, indicating that PZA's anti-inflammatory properties may be critical for its sterilizing activity[32]. Therefore, host modulation may contribute to the mechanism of PZA, and identifying PZA's host target may lead to the development of novel host-directed therapies to improve the treatment of TB.

In this study, we identify the eukaryotic master regulator poly(ADP-ribose) Polymerase 1 (PARP1) as a host target of PZA. PARP1 regulates important cellular functions by generating the post-translational modification poly(ADP-ribose), or PAR, and it is naturally inhibited by NAM and other PZA analogs[20,33–37]. PARP1 drives inflammation, immune and stress responses, including NF-κB transcriptional activity and TNFα signaling[38–43]. We and others therefore hypothesized that PZA might also be a PARP1 inhibitor[44,45] and that PARP1 inhibition may play a mechanistic role in PZA's host-directed activity. Here, we demonstrate that the presence of PARP1 is essential for not only the anti-inflammatory but, more importantly, the antitubercular activities of PZA. We further show that adjunctive use of the FDA-approved PARP1 inhibitor talazoparib (Tp) potently reduces TB lesion size and lung inflammation in mice, potentially by dampening type 1 IFN signaling and neutrophilic inflammation. These findings suggest that PARP1 inhibition may represent an unexplored host-directed therapy strategy for TB that may promote the resolution of lung disease and reduce post-treatment morbidity in TB patients. In addition, PZA may remain clinically useful in the treatment of patients infected with M. tb or related mycobacterial species that show PZA resistance in vitro.

## Results

### PZA is a PARP1 inhibitor

**PZA binds the PARP1 active site.** PZA is a structural analog of the known PARP1 inhibitors nicotinamide (NAM) and benzamide (Fig. 1a)[35]. Langelier et al.[33] recently co-crystalized benzamide adenine dinucleotide (BAD), a non-hydrolyzable NAD+ analog, bound to the ADP-ribosyl transferase (ART) fold of the PARP1 catalytic domain (Fig. 1b, left), and determined that the benzamide moiety forms the basis of the PARP1 binding affinity and enzymatic inhibition of BAD[33]. By structural alignment, we found that PZA is predicted to bind the PARP1 ART fold in the same manner as BAD and that it is predicted to form an additional bond (Phe897) based on our modeling (Fig. 1b, right). Therefore, PZA is predicted to bind the PARP1 catalytic domain similarly to the PARP1 inhibitors NAM and benzamide.

We next confirmed a direct interaction between PZA and PARP1 by differential scanning fluorimetry (DSF) (Fig. 1c). DSF determines the melting temperature ($T_m$) of a protein by incubating it at gradually increasing temperatures in the presence of SYPRO orange, a fluorescent dye that binds exposed hydrophobic residues[46]. Ligand binding increases the thermal stability of a protein, and PZA shifted the PARP1 $T_m$ in a dose-dependent manner similar to NAM (Fig. 1c). The thermal denaturation profile of PARP1 shows multiple peaks representing the unfolding of different PARP1 domains (Supplementary Fig. 1). However, only the 46 °C peak was affected by NAM or PZA binding, as has been previously described for other small molecule inhibitors such as BAD that bind the PARP1 active site[33]. Taken together, our results suggest that PZA directly binds the conserved ART fold of the PARP1 catalytic domain.

**PZA inhibits PARP1 enzymatic activity in macrophages.** The nitrosoguanidine derivative N-methyl-N′-nitro-N-nitrosoguanidine (MNNG) is a potent PARP1 activator that rapidly induces poly-ADP-ribose (PAR) formation in fibroblasts and neurons[47,48]. Relative PARP1 activity can be determined by comparing the levels of target-bound PAR, which appear as high molecular weight smears rather than a defined band on a PAR immunoblot, with a pronounced focus around 116 kDa resulting from PARP1 auto-PARylation[48–50]. We found that MNNG potently activates PAR formation in macrophages, peaking after 10-15 minutes of treatment (Supplementary Fig. 2). Although we were unable to detect PAR formation in M.tb-infected cells, treatment with relevant cytokines (TNFα) or bacterial antigen (LPS) elicited PAR formation similar to MNNG in murine macrophages, primary human peripheral blood mononuclear cells (PBMCs) and HeLa cells (Supplementary Figure 2), indicating that primary and immortalized macrophage-like cells of mouse and human origin can activate PARP1 in response to infection-associated stimuli. We next evaluated PARP inhibition by PZA, NAM, their corresponding acid forms, pyrazinoic acid (POA) and nicotinic acid (NA), or the FDA-approved PARP1/2 inhibitor talazoparib (Tp) in differentiated THP-1 cells stimulated with MNNG (Fig. 2a)[51]. Like NAM, PZA reduced MNNG-induced PAR formation in a dose-dependent manner by up to 80% at concentrations (100–250 μM, or 12.4–31 μg/ml) well within the range of peak serum concentrations reported in patients receiving standard TB therapy (40–80 μg/ml)[15,52]. In contrast, neither POA nor NA significantly inhibited PAR formation, while Tp was

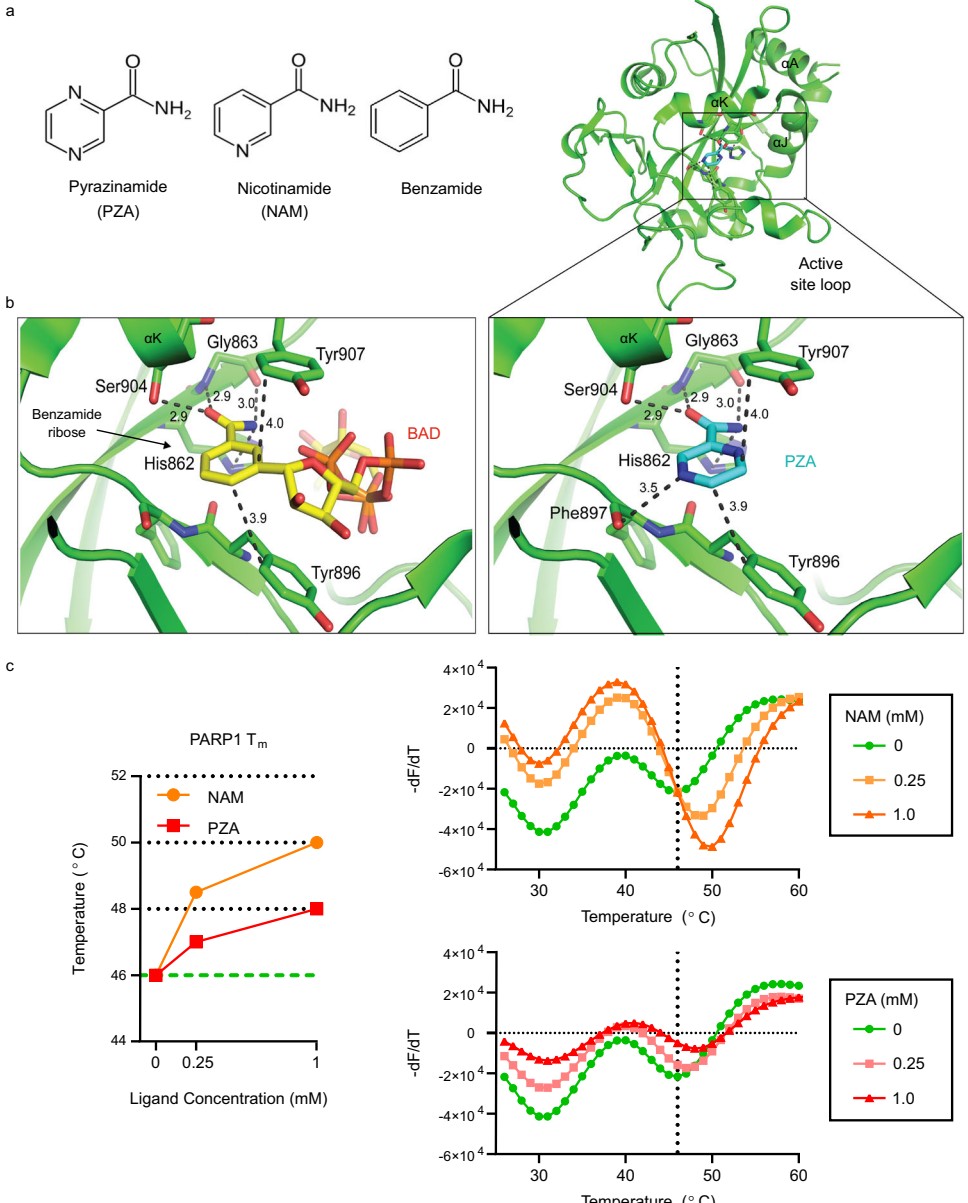

**Fig. 1 | PZA aligns with the PARP1 active site and shifts the PARP1 thermal stability. a** Structures of pyrazinamide (PZA), nicotinamide (NAM) and benzamide. **b** Cartoon of the human PARP1 catalytic domain ART fold (green) co-crystalized with the non-hydrolyzable NAD+ analog benzamide adenine dinucleotide (BAD; left) or aligned with PZA (right). Ligands and key amino acids are drawn in stick representation, and interacting residues and approximate distances (Å) are indicated. Hydroxyl groups are colored red, nitrogen atoms are blue, and the carbon backbone is yellow (BAD) or teal (PZA). Note that PZA is predicted to form one more bond with PARP1 than BAD (Phe897). The PARP1-BAD crystal structure was resolved by Langelier et al.[33] and accessed from the Protein Data Bank (Accession number: 6BHV). **c** PZA directly binds to PARP1 and shifts the PARP1 melting temperature ($T_m$) in a dose-dependent manner. PARP1 $T_m$ (left) and derivative melt curves (right) in the presence of 0.25 mM or 1.0 mM NAM (orange) or PZA (red). Derivative melt curves represent the change in SYPRO orange fluorescence intensity over increasing temperatures of PARP1 alone (green) or with increasing concentrations of NAM (top) or PZA (bottom). The temperature above 40 °C associated with the lowest point in the derivative melt curve (vertical line) was considered the PARP1 $T_m$ (46 °C). Source data are provided as a Source Data file.

the most potent inhibitor of PARP1 activity (Fig. 2b, c). These findings indicate that PZA, but not POA, functions as a PARP1 inhibitor in macrophages.

**PZA inhibits *M.tb*-induced PAR formation in mouse lungs.** To evaluate whether PZA inhibits PARP1 during TB therapy, we compared lung PAR levels in chronically *M.tb*-infected C3HeB/FeJ mice treated with PZA, the PARP inhibitor Tp, the TB antibiotic RIF, or vehicle for two months, the standard PZA treatment duration (Fig. 3a). While PARP activity was low in uninfected mice, PAR levels increased 3.2-fold in the lungs of infected mice before the start of treatment one month after infection (Supplementary Fig. 3a, b). This increase in PAR formation in

response to *M.tb* infection correlated strongly with bacterial burden but not with lung, spleen, or body weights of mice (Supplementary Fig. 3c, d), suggesting that PARP1 activity is enhanced by increasing bacterial burdens and might promote *M.tb* proliferation or persistence in the acute phase of infection.

After 2 months of treatment, PAR levels remained elevated in vehicle-treated mice but were reduced to uninfected levels by both PZA and Tp (Fig. 3b, c, Supplementary Fig. 4a). In contrast, PAR levels in RIF-treated mice were variable but not statistically different from vehicle-treated mice. Since RIF lowered bacterial numbers significantly more than PZA, reduced PAR formation appears to be a specific effect of PZA and independent of any change in bacterial burden (Fig. 3d).

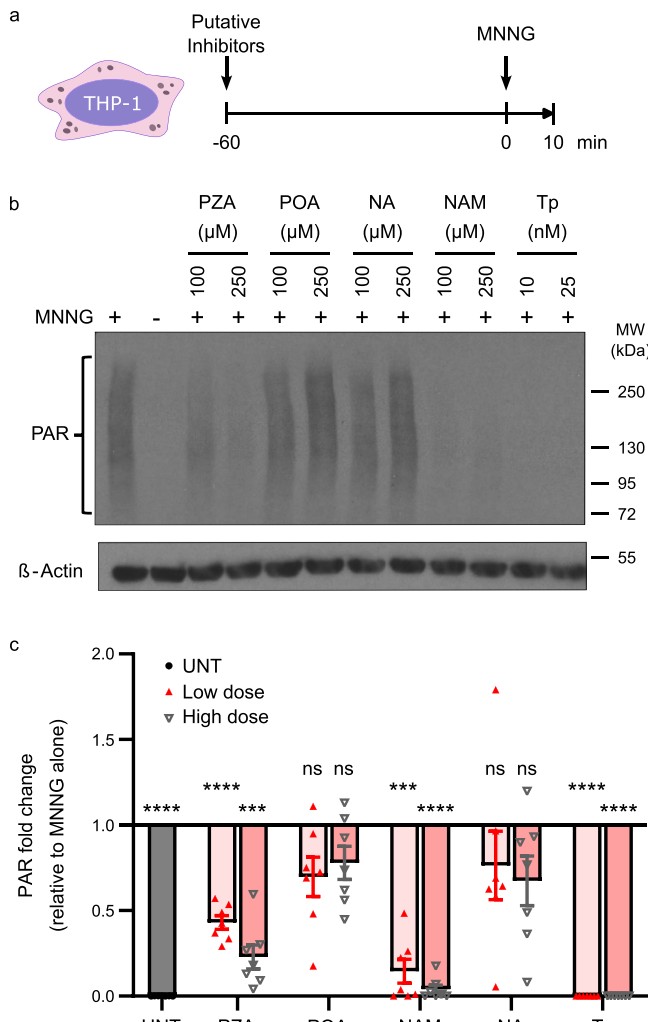

**Fig. 2 | PZA inhibits PARP1 enzymatic activity in macrophages. a** Schematic overview. PMA-differentiated THP-1 cells were incubated with 100 or 250 μM PZA, pyrazinoic acid (POA), NAM or nicotinic acid (NA), or with 10 or 25 nM of the PARP1/2 inhibitor talazoparib (Tp), for 60 min before the addition of PARP1/2 activator MNNG (5 μM). **b** Representative Western blot showing PAR levels (top) or the loading control β-Actin (bottom). **c** Densitometric analysis showing the fold change in ß-Actin-normalized PAR levels relative to cells treated with MNNG alone. UNT is untreated (no MNNG). Error bars represent the SEM from seven independent experiments (individual values indicated by scatter plot). ***$p = 0.0002$ (PZA high dose) or 0.0001 (NAM low dose); ****$p < 0.0001$; ns not significant; by repeated measures (RM) one-way ANOVA with Dunnett's multiple comparisons test. PAR levels in cells treated with 250 μM PZA (22.9%) were significantly lower than with 100 μM PZA (43.1%) but not statistically different from cells treated with 100 μM NAM (14.6%). Source data are provided as a Source Data file.

These results support the conclusion that PZA is a functional PARP1 inhibitor in vivo at clinically relevant concentrations that suppresses PAR formation to baseline levels during TB chemotherapy.

Uniformly low bacterial numbers despite wide-ranging PAR levels in RIF-treated mice further demonstrate 1) that the bactericidal efficacy of RIF is independent of PARP1 activity and 2) that bacterial killing alone is insufficient to reduce PAR formation in infected mice (Supplementary Fig. 4b, c). Tp-treated mice on the other hand tended to have comparable or higher CFU than vehicle-treated mice, indicating that PARP inhibition alone does not improve *M.tb* clearance. In fact, even though there were no significant correlations in individual treatment groups, PAR levels in mice that did not receive antibiotics (vehicle and Tp-treated) were trending toward a weak inverse correlation with bacterial burden (Supplementary Fig. 4c–e). Together, our

findings suggest that PARP1 activation may promote bacterial expansion in the acute phase of infection (Supplementary Fig. 3c) but contribute to bacterial containment during the chronic phase of infection (Supplementary Fig. 4e), indicating that PARP1 inhibition without effective antibiotics may antagonize bacterial clearance.

In contrast to all other groups, PZA-treated mice followed a unique pattern that illustrates the complexity of its dual host-directed and antibiotic activities. PARP1 inhibition and bacterial killing coincided in 50% (4/8) of PZA-treated mice, while bactericidal activity without PARP1 inhibition was only evident in a single mouse (12.5%). Interestingly, PZA was completely ineffective in one of the two animals in which PZA did not inhibit PARP1, further strengthening the correlation between PARP inhibition and PZA efficacy. Two additional PZA-treated mice (25%) had low PAR levels but bacterial burdens higher than the vehicle average, supporting that PARP1 inhibition alone is not sufficient to clear the infection. These findings suggest that PARP1 inhibition is not required for but potentiates the bactericidal efficacy of PZA.

### Adjunctive PARP1 inhibition improves TB lung disease
**PARP1 inhibition reduces lung inflammation and lesion size in mice.**
PARP1 is a pro-inflammatory master regulator and a driver of acute (LPS-induced) and chronic (asthma) lung inflammation, and its inhibition has been shown to ameliorate disease severity in conditions ranging from septic shock to hepatitis[41,53,54]. To gain insight into the functional consequences of PARP1 inhibition during TB therapy, we next evaluated disease progression, lung inflammation, and immune responses in chronically *M.tb*-infected C3HeB/FeJ mice treated with PZA or Tp alone or in combination with RIF for two months (Fig. 4; Supplementary Figs. 5–8). In culture, Tp has minimal bactericidal activity in the same range as its solvent DMSO alone only at the highest concentration tested (128 μg/ml) and no discernable inhibitory effects at eight times the concentration used to treat mice (Supplementary Table 1). Since Tp (PARP inhibitor but no bactericidal activity) predominantly influences host responses and RIF (bactericidal activity but no PARP inhibition; Fig. 3c, d) primarily exerts antimycobacterial effects, the combination of Tp and RIF most closely mimics the dual host-directed and antibiotic functions of PZA. Following 2 months of treatment, the bacterial burdens showed that, as expected, Tp monotherapy ($8.2 \pm 0.2$ log₁₀ CFU, $p = 0.45$ vs. vehicle) had no bactericidal activity compared with vehicle alone ($7.5 \pm 0.2$ log₁₀ CFU) while PZA monotherapy showed modest bactericidal activity ($6.0 \pm 0.4$ log₁₀ CFU, $p = 0.0006$ vs. vehicle) that was statistically inferior to RIF monotherapy ($4.3 \pm 0.1$ log₁₀ CFU, $p < 0.0001$ vs. vehicle or PZA). Consistent with the protective role of PARP1 during the chronic phase of infection, the addition of Tp to RIF ($5.5 \pm 0.2$ log₁₀ CFU for RIF + Tp, $p < 0.0001$ vs. vehicle) slightly antagonized the CFU benefit of RIF alone ($p = 0.0067$ vs. RIF), while the addition of PZA ($0.5 \pm 0.3$ log₁₀ CFU for RIF + PZA, $p < 0.0001$ vs. vehicle or RIF) significantly augmented the bacterial killing observed with RIF alone (Fig. 4b).

Although the above lung CFU counts were unsurprising, we observed unexpected lung pathology changes among the treatment groups. Three months after infection, vehicle-treated mice displayed hallmark features of TB lung disease, including necrotic and non-necrotic granulomas and extensive cellular infiltration (Fig. 4c; Supplementary Fig. 5). While these pathological features were reduced in animals receiving PZA or RIF monotherapy, the most striking improvements were observed in mice receiving RIF + Tp. Histological analysis revealed that mice treated with RIF + Tp indeed had the lowest amount of inflammation and lung involvement of all groups (Fig. 4d, e), despite harboring significantly more bacilli than mice treated with RIF or RIF + PZA (Fig. 4b). In fact, the addition of Tp to RIF significantly reduced both lung inflammation (Fig. 4d) and lung involvement (Fig. 4e) along with a 2-log reduction in bacterial burden compared to vehicle-treated mice ($p < 0.0001$). Correlating histological findings

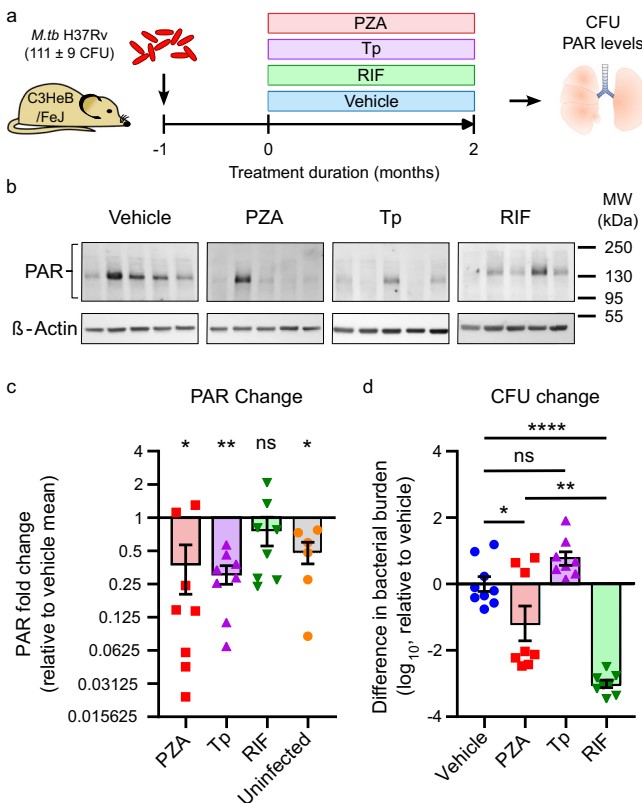

**Fig. 3 | PZA reverses TB-induced PARP1 activation in mouse lungs. a** Schematic study overview. Female C3HeB/FeJ mice were aerosol infected with *M.tb* H37Rv (implantation: 111 ± 9 CFU) and administered PZA (150 mg/kg), the PARP inhibitor talazoparib (Tp; 0.5 mg/kg) or vehicle (1.97% DMSO in 0.5% CMC), alone or in combination with the TB antibiotic RIF (10 mg/kg), 5 days/week for 2 months starting 1 month post infection. Bacterial burden and PAR levels were assessed before and after treatment and in age-matched uninfected control mice. **b** Representative Western blots showing PAR levels (top) or the loading control β-Actin (bottom) in mouse lungs after 2 months of treatment. Each lane represents an individual mouse. **c** Densitometric analysis showing the change in β-Actin-normalized lung PAR intensity relative to the mean in vehicle-treated mice. *n* = 6 (uninfected) or 8 (all other groups). Each symbol represents an individual mouse and bars the group mean ± SEM. ns not significant (RIF vs. vehicle); *$p$ = 0.0159 (PZA vs. vehicle) or 0.0433 (uninfected vs. vehicle); **$p$ = 0.0071 (Tp vs. vehicle) by two-way ANOVA with uncorrected Fisher's least significant difference (LSD) test. **d** Corresponding difference in lung bacterial burden following treatments compared to vehicle-treated mice. *n* = 9 (vehicle) or 8 (all other groups). Each symbol represents an individual mouse and bars the group mean ± SEM. ns not significant (Tp vs. vehicle); *$p$ = 0.0425 (PZA vs. vehicle); **$p$ = 0.0013 (PZA vs. RIF); ****$p$ < 0.0001 (RIF vs. vehicle) by one-way ANOVA with Tukey's multiple comparisons test. Mice treated with PZA or Tp had significantly reduced lung PAR levels comparable with uninfected mice. PAR levels in RIF-treated mice were not statistically different from vehicle-treated mice, even though the bacterial burden was significantly more reduced by RIF than by PZA or Tp. Source data are provided as a Source Data file.

with the corresponding bacterial burden indicated that both PZA and Tp lowered inflammation and lung involvement more than expected (Supplementary Fig. 6a). However, while PZA appeared to consolidate TB lesions (reduced frequency, larger size), Tp had the opposite effect (increased frequency, smaller size); in contrast, all of the other groups showed no consistent trend (Supplementary Fig. 6b). We next quantified fibroblasts, macrophages, neutrophils, CD4+ and CD8+ T-cells to determine if changes in immune cell infiltration might explain the observed lung pathology patterns. The presence of fibroblasts in the lung mirrored the pattern of inflammatory changes and was most potently reduced by the combination RIF + Tp, indicating that

adjunctive PARP inhibition may have anti-fibrotic effects in TB-infected lungs (Fig. 4f). In addition, Tp potently reduced CD4+ T-cell, and to a lesser extent CD8+ T-cell and macrophage, staining in infected lungs but only reduced neutrophil numbers in the lung or in granulomas when combined with RIF (Supplementary Fig. 6c–e). This is consistent with the emerging understanding of the role of PARPs in regulating innate immune responses, especially in the recruitment and function of neutrophils[55,56]. Together, our data suggest that PARP inhibition may accelerate the resolution of TB lung disease when used in combination with effective antibiotics by reducing neutrophil recruitment, CD4+ T cell responses, and lung fibrosis, thus antagonizing granuloma formation or maintenance.

## PARP1 contributes to PZA's anti-inflammatory host effects

**PARP1 inhibition dampens TB immune responses in mice.** PARP1 is a transcriptional regulator and NF-κB co-activator that enhances the production of inflammatory mediators fundamental for the host response to TB, including TNFα, IFNγ, and IL-1β[39,40,42,43]. Manca et al.[22] demonstrated that PZA suppresses these responses at a transcriptional level in a pattern suggestive of NF-κB modulation but were unable to identify PZA's host target[22]. To evaluate whether PZA's anti-inflammatory effects could be mediated by PARP1 inhibition, we next compared lung levels of TNFα, IFNγ, and IL-1β in C3HeB/FeJ mice treated with PZA (PARP inhibitor and bactericidal activity), the PARP inhibitor Tp (no bactericidal activity) or the antibiotic RIF (no PARP-inhibitory activity) alone or in combination (Fig. 4a). Paralleling our histological observations, we found that both Tp and PZA potently reduced IFNγ and to a lesser extent IL-1β levels while TNFα inhibition was only observed in the presence of antibiotics (PZA or RIF; Supplementary Fig. 8a). RIF treatment alone also decreased the levels of the three cytokines, suggesting that control of bacterial burden is a primary contributor to the reduction of lung cytokines. We further teased apart the relative contributions of PARP inhibition and bacterial killing on the anti-inflammatory properties of PZA at the transcriptional level by qPCR (Supplementary Fig. 8b). Interestingly, the expression of *Tnfα* and inducible Nitric Oxide Synthase (*iNOS*) was potently inhibited by combined PARP1 inhibition and bacterial killing (PZA or RIF + Tp), while bacterial killing alone (RIF) or PARP1 inhibition alone (Tp) had only modest effects. In addition, PARP1 inhibition had remarkable effects on interferon (IFN) signaling: while Type I IFN (*Ifnb*) expression was modestly reduced by RIF or Tp alone and potently by PZA or RIF + Tp, the IFN-simulated genes interferon-induced protein with tetratricopeptide repeats 1 (*Ifit1*) and 3 (*Ifit3*) were significantly inhibited by Tp, PZA or RIF + Tp while RIF alone had minimal effects on their expression. Recent blood transcriptomic analysis identified IFIT1 and IFIT3 as two of the few main risk factors predictive for TB progression in humans[57]. In contrast, the expression of monocyte chemoattractant protein-1 (*Mcp-1*) and C–X–C motif chemokine receptor 5 (*Cxcr5*) were repressed or stimulated, respectively, in direct correlation with bacterial burden and showed minimal effects of PARP1 inhibition. Together, our findings suggest that the potent anti-inflammatory properties of PZA result from its combined ability to reduce bacterial burdens (lowering *Mcp-1* and *Il-1b* expression, increasing *Cxcr5* expression) and inhibit PARP1 (lowering *Tnfα*, *iNos*, *Ifnγ*, *Ifnb*, *Ifit1* and *Ifit3* expression), leading to reduced levels of pro-inflammatory mediators (IFNγ, IL-1β, and TNFα) and type I IFN signaling in the lung[58]. These patterns support the conclusion that PARP1 inhibition appears to contribute to PZA's host-directed activity.

**PZA modulates TB immune responses in a predominantly PARP1-dependent manner.** To more definitively ascertain whether the host effects of PZA are mediated by PARP1 inhibition, we compared PZA's immune-modulatory activity in 129 S1-SvlmJ (WT) and PARP1-deficient 129S-*Parp1^tm1Zqw*/J (PARP1−/−) mice (Fig. 5; Supplementary Fig. 9). PZA is a prodrug converted to its active form pyrazinoic acid (POA) by the

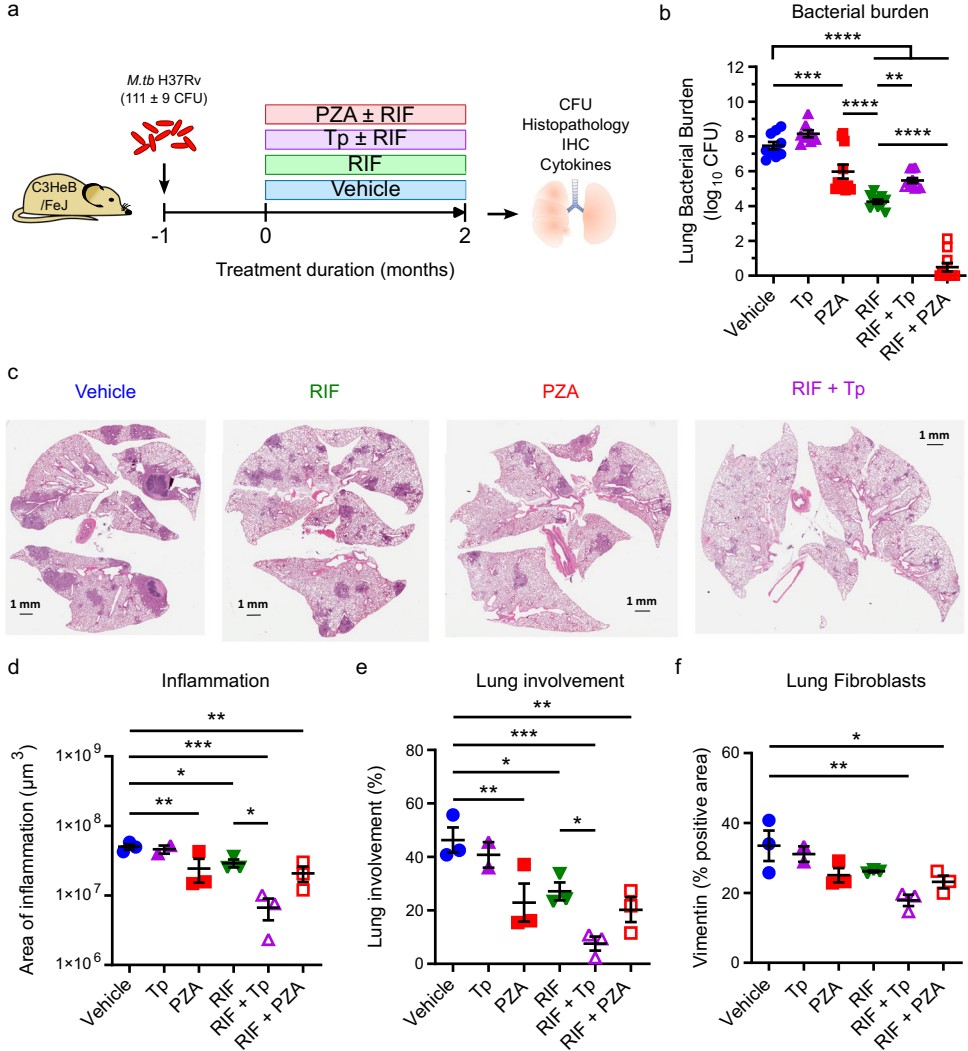

**Fig. 4 | Adjunctive PARP inhibition reduces TB lung inflammation.** Lung histopathology of *M.tb*-infected female C3HeB/FeJ mice (implantation: 111 ± 9 CFU) 3 months post infection following 2 months of treatment with PZA (150 mg/kg), talazoparib (Tp; 0.5 mg/kg) or vehicle (1.97% DMSO in 0.5% CMC), alone or in combination with RIF (10 mg/kg). **a** Schematic overview. **b** Lung bacterial burden at the end of treatment. $n = 8$ (Tp), 10 (vehicle, RIF, RIF + PZA) or 11 (PZA, RIF + Tp). Statistical differences were determined by one-way ANOVA with Šidák's multiple comparisons test. **p = 0.0067; ***p = 0.0006; ****p < 0.0001. **c** Representative H&E-stained lung sections. **d**–**f** Quantified areas of lung inflammation (**d**), % lung involvement (**e**), or quantified vimentin-positive area indicative of fibroblasts expressed as a percent of total lung area (**f**). Each symbol represents an individual mouse, with mean ± SEM indicated. $n = 2$ (Tp) or 3 (all other groups). Statistical differences were determined by one-way ANOVA with uncorrected Fisher's LSD test (**d**, **e**) or Dunnett's multiple comparisons test (**f**) with a single pooled variance. *p < 0.05; **p < 0.01; ***p < 0.001; ****p < 0.0001; exact p values are provided in the Source Data file. Histopathology and IHC analyses were performed on randomized and coded slides by a veterinary pathologist blinded to experimental design. Adjunctive PARP inhibition reduced lung inflammation independently of bacterial burden. Source data are provided as a Source Data file.

mycobacterial pyrazinamidase PncA, and mutations in the *pncA* gene are known to render *M.tb* resistant to PZA[59,60]. Since we were solely focused on PZA's host-directed activity, we utilized a PZA-resistant *M.tb* mutant (H37RvΔ*pncA* (A146V)) to eliminate any direct bactericidal activity and confounding differences in bacterial burden following PZA treatment. Even though PZA did not significantly reduce the bacterial burden in WT or PARP1$^{-/-}$ mice after 2 months of treatment as expected with the resistant strain (Fig. 5b), it uniformly reduced the levels of cytokines and chemokines in WT mice (Fig. 5c, d; Supplementary Fig. 9). Remarkably, these immune-modulatory effects were largely absent in PARP1$^{-/-}$ mice. This pattern was most pronounced for IFNγ, IL-1β, IL-12, and TNFα, the monocyte chemoattractant MCP-1, and the T-cell chemoattractant RANTES (Fig. 5d). In contrast, IL-10, IL-6 and the neutrophil chemoattractant KC appeared modestly suppressed by PZA in both WT and PARP1$^{-/-}$ mice, suggesting that PZA may also weakly modulate PARP1-independent immune mechanisms, while the

neutrophil chemoattractant LIX was unaffected in either strain. These data confirm a specific host-directed effect of PZA that is independent of its bactericidal activity but largely dependent on the presence of PARP1.

## PARP1 contributes to the antitubercular efficacy of PZA
**PZA's bactericidal efficacy is impaired in PARP1$^{-/-}$ mice.** Having identified PARP1 as a host target of PZA and demonstrated that PARP1 is required for PZA's host-directed activity, we sought to determine if PARP1 is required for the antitubercular activity of PZA. To address the relative impact of the host-directed PARP1-inhibitory activity of PZA, we evaluated the efficacy of standard PZA treatment in WT and PARP1$^{-/-}$ mice infected with PZA-susceptible *M.tb* H37Rv (Fig. 6). While PZA reduced bacterial numbers in both WT and PARP1$^{-/-}$ mice, the bacterial burden was significantly higher in PARP1$^{-/-}$ than in WT mice at the end of treatment (5.2 ± 0.2 vs. 4.4 ± 0.2 log$_{10}$ CFU, respectively; $p = 0.0493$)

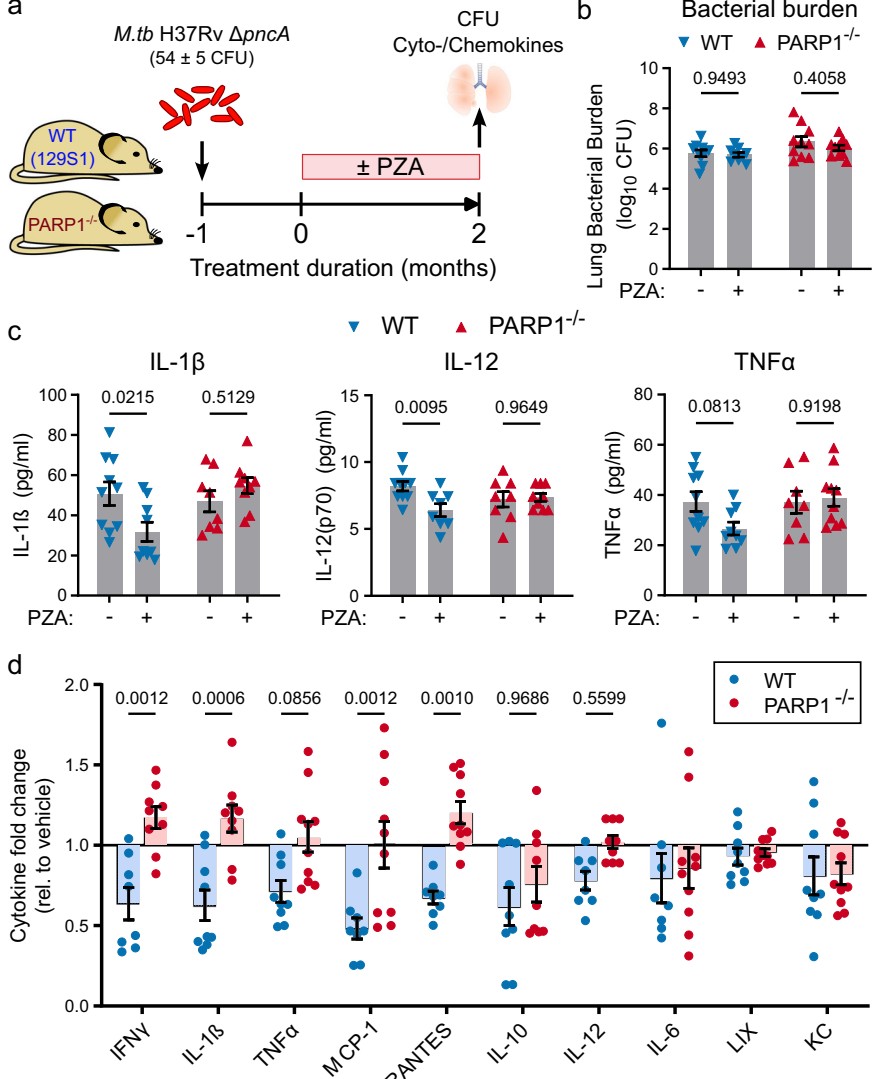

**Fig. 5 | PZA's anti-inflammatory effects are PARP1-dependent. a** Schematic overview. Male and female PARP1-null (PARP1$^{-/-}$) or 129S1 (WT) mice were aerosols infected with the PZA-resistant *M.tb* H37Rv Δ*pncA* (A146V) mutant (implantation: 54 ± 5 CFU). Starting one month after infection, half of the mice were administered PZA (150 mg/kg) 5 days/week for 2 months before lung bacterial burden and cyto-/chemokine levels (Luminex multiplex assay) were determined. *n* = 9 (PZA-treated WT) or 10 (all other groups). **b, c** Bacterial burden (**b**) and cytokine/chemokine concentrations (**c**) in untreated (−) or PZA-treated (+) mice

at the end of treatment. **d** Change in cyto-/chemokine levels in PZA-treated mice relative to the levels in untreated mice of the same strain. Values below 1 indicate levels that are lower in the lungs of PZA-treated than untreated mice. Each symbol represents an individual mouse, with mean ± SEM indicated. Statistical differences between groups were determined by two-way ANOVA with Šidák's multiple comparisons test. *p* Values for all relevant comparisons are indicated in the figure. PZA reduced lung cyto- and chemokine levels in WT but not in PARP1$^{-/-}$ mice and independently of bacterial burden. Source data are provided as a Source Data file.

even though the bacterial burden in untreated mice was comparable between strains at all time points before and after treatment (Fig. 6b–d; Supplementary Table 2). In contrast to our findings with the PZA-resistant *M.tb* mutant (Fig. 5), PZA treatment reduced proinflammatory cytokine levels in both WT and PARP1$^{-/-}$ mice (Supplementary Fig. 10). However, the magnitude of these responses was generally lower in PARP1$^{-/-}$ mice, further supporting our hypothesis that PZA's anti-inflammatory effects result from its combined PARP-inhibitory and bactericidal activities. Together, these results implicate PARP1 as a key factor in the mechanism of PZA that contributes not only to immune modulation but also to bacterial clearance (Fig. 6e). While we do not rule out additional PARP1-independent immune mechanisms targeted by PZA, our findings support that modulation of the host environment is a central component of PZA's sterilizing efficacy.

## Discussion

PZA is a puzzling antibiotic with an unresolved mechanism of action. While it significantly shortens the time for successful TB treatment in humans and shows powerful sterilizing activity in animal models, its bactericidal activity is negligible in humans and modest in animal models[15,16,32]. Though earlier studies indicated that PZA has host-directed, anti-inflammatory effects, PZA's host target has remained elusive[21,22]. Here, we identify the proinflammatory master regulator PARP1 as the host target of PZA and demonstrate that PZA's anti-inflammatory effects are largely absent in mice lacking PARP1. Additionally, PZA's bactericidal efficacy was attenuated in PARP1-deficient mice, suggesting that immune modulation may be an integral component of PZA's sterilizing activity. Since PARP1 is a prominent driver of inflammation in human disease, PARP1 inhibition may be the missing piece to solve the puzzle of PZA's mechanism of action.

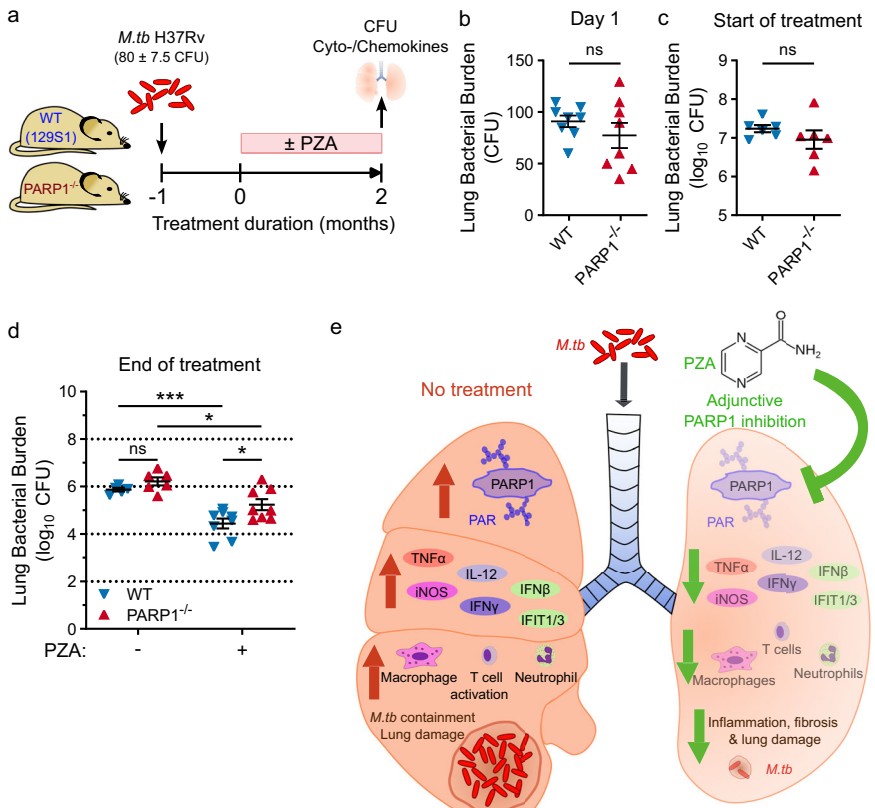

**Fig. 6 | PZA has reduced efficacy in PARP1⁻/⁻ mice. a** Study outline to compare the efficacy of PZA in WT and PARP1⁻/⁻ mice. Male and female PARP1-null (PARP1⁻/⁻) or 129S1 (WT) mice were aerosols infected with *M.tb* H37Rv (implantation: 80 ± 7.5 CFU). Starting one month after infection, half of the mice were administered PZA (150 mg/kg) 5 days/week for 2 months. Lung bacterial burden was assessed at the beginning and end of treatment. **b**–**d** Lung bacterial burden on the day after infection (**b**; *n* = 8), at the start of treatment (**c**; *n* = 6), and in untreated (−) or PZA-treated (+) mice at the end of treatment (**d**; *n* = 5 (untreated WT), 6 (untreated PARP1⁻/⁻) or 8 (PZA-treated)). Each symbol represents an individual mouse, with mean ± SEM indicated. Statistical differences between groups were determined by two-tailed unpaired *t*-test (**b**, **c**) or two-way ANOVA with Šidák's multiple comparisons test (**d**). *$p$ = 0.0135 (PARP1⁻/⁻ untreated vs. PZA) or 0.0361 (PZA: WT vs.

PARP1⁻/⁻); ***$p$ = 0.0005; ns not significant. After 2 months of PZA treatment, significantly more bacilli remained in the lungs of PARP1⁻/⁻ than of WT mice (5.234 vs. 4.437 log₁₀ CFU, respectively). **e** Proposed model: PARP1 inhibition is a key component of PZA's mechanism of action in TB therapy. Without treatment (left), *M.tb* infection activates PARP1 which promotes the production of inflammatory mediators, immune cell activation and *M.tb* containment at the cost of lung damage. Inhibiting PARP1 during TB therapy (right), by PZA or adjunctive PARP inhibition, can accelerate bacterial clearance and the resolution of lung damage while dampening inflammation. However, PARP1 inhibition without adequate antibiotics may impair bacterial containment. Source data are provided as a Source Data file.

Although it is widely accepted that the bactericidal activity of PZA is strongly influenced by the host environment, the clinical relevance of PZA's host-directed activity has remained a topic of debate. On the one hand, the fact that mutations in non-essential mycobacterial genes, like *pncA*, are sufficient to cause PZA resistance supports the argument that the mechanism of action of PZA is antimicrobial[60,61]. On the other hand, PZA is known to reduce proinflammatory cytokines, and decreasing such mediators has repeatedly been shown to make *M.tb* more susceptible to drug- or immune-mediated killing mechanisms and accelerate bacterial clearance[21,22,24,62–65]. Our demonstration in this study that PZA-mediated reductions in proinflammatory cytokines are PARP1-dependent provides molecular insight and adds compelling further evidence in support of a host-directed mechanism for PZA.

Almeida and colleagues[30] attempted to tease apart the antibacterial versus host-directed activity of PZA and concluded that even though the bactericidal value of PZA was either lost (in combination with RIF) or greatly reduced (in combination with rifapentine) in mice lacking a functional immune system, host effects are unlikely to comprise a major component since PZA was ineffective against the PZA-resistant species *M. bovis*[30]. Interestingly, Simmons et al.[31] reported that PZA restricts the growth of attenuated (*M. bovis*-BCG) but not virulent (WT *M. bovis*; *M.tb* H37Rv *ΔpncA*) mycobacteria in macrophages, even though all of these strains are resistant to direct killing by

PZA in vitro. This suggests that PZA's host-directed activity may suffice to promote the clearance of weakened but not fully virulent mycobacteria[31]. Intriguingly, PZA induces extensive metabolic changes in mycobacteria, including in virulence factor synthesis, which could conceivably weaken even virulent *M.tb* enough to promote host-mediated bacterial clearance[66]. We similarly observed no change in bacterial burden following PZA treatment in mice infected with *M.tb* H37Rv *ΔpncA* despite PZA's immune-modulatory activity, indicating that PZA's host effects alone are not sufficient to induce the clearance of virulent *M.tb*. However, our observations in PARP1⁻/⁻ mice suggest that the potency of PZA may be the result of a unique combination of bactericidal and host-directed activities. As Almeida et al.[30] clearly demonstrated, when either of these two elements is missing the efficacy of PZA is greatly reduced, even when bacterial replication is inhibited[30].

Our regression analyses suggest that PARP1 activity may promote *M.tb* infection during the acute phase (Supplementary Fig. 3c) but enhance *M.tb* containment during the chronic phase of infection (Supplementary Fig. 4e). Consequently, PARP1 inhibition without adequate antibiotic activity (Tp alone or in combination with RIF monotherapy) was associated with increased bacterial burdens but paradoxically reduced inflammation, lesion size and immune cell infiltration (Fig. 4, Supplementary Fig. 6). At the cellular level, PARP1

inhibition most prominently reduced fibroblast, macrophage and T-cell numbers in infected mouse lungs (Fig. 4, Supplementary Fig. 6) which are fundamental components of the hallmark TB granuloma, an immune structure that limits the spread of *M.tb* but may inadvertently hinder immune- or antibiotic-mediated bacterial killing[5,24,65]. This pattern is reminiscent of the dual role of TNFα in TB, which is critical for bacterial control but also drives tissue destruction and impairs the bactericidal efficacy of standard TB therapy[23,24,67]. Together with the observation that adjunctive PARP1 inhibition suppressed TNFα, INFγ and iNOS expression, our findings implicate PARP1 as a master regulator of the double-edged proinflammatory TB immune response.

In addition, PARP1 appears to regulate protective T-cell responses since PARP1 inhibition lowered CD4+ T-cell frequencies as well as CXCR5 expression in mouse lungs (Supplementary Fig. 6 and 8b). Since CD4+ CXCR5+ T-cells have been shown to promote protective immunity against TB[58], these observations offer a potential explanation for the reported antagonism between PZA and RIF[32] and impaired TB containment associated with PARP1 inhibition in our study. On the other hand, C3HeB/FeJ mice displaying the least lung pathology (RIF + Tp, RIF + PZA) had dramatic reductions in both lung and granuloma neutrophils. PARP1 inhibition also potently suppressed the expression of IFNβ and the type I IFN-inducible genes IFIT1 and IFIT3, which were recently identified as biomarkers predictive of active TB[57]. Type I IFNs drive TB susceptibility and pathogenesis by inducing neutrophilic inflammation which promotes disease progression[68]. Our findings implicate Type I IFN signaling as a target of adjunctive PARP1 inhibition and a potential mechanism behind the dramatically improved lung pathology observed in RIF + Tp-treated mice. Our results collectively demonstrate that adjunctive PARP1 inhibition has the potential to accelerate TB lung healing but should be further optimized to minimize the risk of impeding bacterial clearance and evaluated in the context of standard TB therapy. Importantly, aberrant type I IFN and neutrophil responses are hallmarks of non-resolving pulmonary TB despite antibiotic therapy[68], and these patients may especially benefit from adjunctive PARP1 inhibition.

This work has several clinical ramifications. First, our finding that adjunctive PARP inhibition with Tp dampened TB immune responses and potently reduced pathology in mouse lungs suggests that the addition of non-antimicrobial PARP1 inhibitors as adjunctive therapy may benefit the treatment of TB by reducing lung disease. Considering the fact that new regimens, such as bedaquiline-pretomanid-linezolid (BPaL), that are free of the traditional first-line anti-TB drugs are now used for the treatment of drug-resistant TB, an argument can be made that PARP1 inhibition may add a measure prevention against tissue damage[69]. Excessive, nonproductive inflammation and tissue damage associated with TB infection can greatly impair the quality of life of TB survivors, up to half of whom suffer from persistent or progressive lung dysfunction and remain at high risk of developing chronic lung disease even after being cured of the infection[6,70-74]. PARP1 is a known driver of chronic inflammation, and our data indicate that PARP1 inhibition in combination with an effective regimen of TB antibiotics may prevent TB-induced lung damage when used as adjunctive therapy.

A second clinically relevant aspect of our work pivots on the use PZA in patients infected with *M. bovis*, *M. kansasii* (both species which are naturally resistant to PZA in vitro) or *M. tb* strains known to be PZA-resistant. Current WHO and expert panel guidelines consider PZA to be an effective anti-TB drug only when drug-susceptibility patterns confirm susceptibility[75]. Despite this, some clinical studies have shown that PZA-treated patients experience equivalent or comparable levels of treatment success regardless of whether they are infected with PZA-resistant or -susceptible strains[76-78]. Intriguingly, the bactericidal efficacy of PZA coincided with reduced PARP1 activity in 4/5 mice (80%), supporting our hypothesis that PARP1 inhibition is integral to the antimycobacterial activity of PZA, but since PZA also reduced PAR

levels but not CFU (2/8) or CFU but not PAR levels (1/8) in a small number of mice suggests that the bactericidal and host-direct effects of PZA are independent but complementary mechanisms. Remarkably, PZA on its own has no bactericidal activity in TB patients infected with fully drug-susceptible *M.tb* but uniquely reduces the size and inflammation of highly inflamed lesions[32], suggesting that host-directed rather than bactericidal effects confer the therapeutic benefit of PZA and that infection with a PZA-resistant strain may more accurately portray PZA's activity in humans. Using a PZA-resistant mutant, our results clearly demonstrate that PZA lowers inflammatory cytokine responses in a PARP1-dependent manner even in the absence of bacterial killing. PZA or adjunctive PARP1 inhibition thus may indeed offer a clinical benefit even against mycobacteria that show microbiologic or mutational evidence of resistance to the drug.

Our study has several limitations, including our inability to detect PARP activity in *M.tb*-infected cells (Supplementary Fig. 2d) which hindered the study of PZA in vitro (despite numerous attempts using virulent or attenuated strains, including *M.bovis* BCG, *H37Ra* and *M. smegmatis*; various MOIs; with or without IFNγ priming; and time-points ranging from 30 min to 48 h post-infection). In addition, we observed much narrower PAR smears focused around the site of PARP1 auto-PARylation in mouse lungs than in cytokine- or MNNG-simulated cells that may reflect differences in the strength of PARP1 activation in a heterogenous tissue[50], result from the rapid degradation of ADP-ribose polymers by enzymes and proteases during tissue processing, or both. Lastly, while we speculate that adjunctive PARP1 inhibition may benefit the treatment of drug-resistant *M.tb*, we did not investigate the effectiveness of PARP inhibitors in MDR/XDR TB or other PZA-resistant strains.

If PARP1 were PZA's only host target, we would expect PARP1 knockout to mimic PZA treatment of WT mice. However, lack of PARP1 also has many effects on cell-mediated and humoral immune responses that may affect early TB responses differently than simply inhibiting PARP1 after the infection is already established[39]. Despite these known developmental immune deficiencies, PARP1$^{-/-}$ mice were nonetheless capable of containing the infection and mounting immune responses that were largely comparable to WT mice, indicating that other processes, including other PARPs, can compensate for the absence of PARP1 to restore effective host responses. Since the ART fold and NAM binding pocket are highly conserved among members of the PARP superfamily, small molecules such as Tp and PZA may inhibit multiple PARP family members, in particular PARPs 1-6[79]. Observed differences could thus also be due to the broad-spectrum inhibition of multiple PARPs. However, our observation that cytokine levels in PARP1$^{-/-}$ mice were largely unaffected by PZA implies PARP1-compensatory mechanisms are not targets of PZA and are unlikely to contribute to PZA's mechanism of action. Importantly, our data in Fig. 6 suggests that without its ability to modulate PARP1-dependent responses the bactericidal efficacy of PZA is diminished. Since PZA has negligible bactericidal but potent anti-inflammatory activity in human TB patients[32], our data collectively suggest that PARP1 inhibition may be a key component of PZA's mechanism of action underlying its unique treatment-shortening ability in TB therapy.

In summary, this study indicates that the PARP1-dependent, host-directed activity of PZA may comprise an integral component of its sterilizing mechanism by altering tissue pathology and pro-inflammatory cytokine release. Our work suggests that non-antibacterial PARP1 inhibitors may have value as adjunctive agents in the treatment of TB and that PZA may confer a benefit even in the setting of microbiologic PZA resistance.

## Methods
### Ethics statement
All experiments with *Mycobacterium tuberculosis* were carried out in Institutional Biosafety Committee-approved BSL3 and ABSL3 facilities

at The Johns Hopkins University School of Medicine using recommended positive-pressure air respirators and protective equipment. Experimental procedures involving live animals were carried out as described in protocols #M022M466 and #MO22M134 approved by the Institutional Animal Care and Use Committee (IACUC) at the Johns Hopkins University School of Medicine. Animals used for the experiments were maintained on 12:12 h light/dark cycle with free access to food and water. Aerosol infection, accomplished by placing the mice in a Glas-Col aerosolization instrument, did not require anesthesia. The mice were free to move about within the aerosol chamber and experienced minimal pain, distress, or discomfort. Mice were briefly restrained for oral gavage. Some *M.tb* infected mice tended to lose weight, and were sluggish in disposition. Mice that became moribund prior to the end of the required experimental time period were sacrificed immediately, except for in survival studies. At the end of the experimental time period, mice were humanely sacrificed by procedures consistent with AVMA guidelines.

### Molecular docking studies
PZA was manually docked into the crystal structure of PARP1 (PARP1 CAT ΔHD) (PDB Accession number: 6BHV) in *COOT* software based on the information published by Langelier et al.[33] of the ligand benzamide adenine dinucleotide (BAD) bound to PARP1 CAT ΔHD in crystal structure[33,80]. Figures were prepared using PyMOL Molecular Graphics System, Version 1.5.0.4 (Schrödinger, LLC).

### PARP1 thermal shift assay
High-purity recombinant human PARP1 protein (UniProt ID: P09874) in 100 mM Tris-HCl (pH 7.5) containing 14 mM β-mercapthoethanol, 0.5 mM EDTA, 0.5 mM PMSF, and 10% glycerol was purchased from Enzo Life Sciences. 5000x of SYPRO™ Orange (Invitrogen) was diluted to 50x in water. For the thermal shift assay, 4 μg of PARP1 protein and 3x of SYPRO™ Orange were combined in a 0.1 ml MicroAmp Fast 96-well reaction plate (Applied Biosystems) with 0, 0.25, or 1.0 mM PZA or NAM, respectively, in a total volume of 50 μl per well. Fluorescence data was collected on a StepOnePlus Real-Time PCR System using the StepOne software v2.3 (Applied Biosystems). ROX (SYPRO Orange) was selected as the reporter dye, and no dye was selected as a passive reference. Temperature was gradually increased from 25 to 60 °C and held for 1 min/degree. Melting temperature (Tm) and differential fluorescence ($-dF/dT$) values were calculated by fitting the data to the Sigmoidal dose-response (variable slope) equation in GraphPad Prism version 5.01 for Windows (GraphPad).

### Bacterial strains
*M.tb* strain H37Rv was obtained from the Johns Hopkins Center for Tuberculosis Research. The PZA-resistant strain H37RvΔ*pncA* (A146V), containing a single point mutation in the pyrazinamidase gene *pncA*, was a generous gift from Dr. Eric Nuermberger[81]. Mycobacteria were grown to an optical density at 600 nm of approximately 1.0 in Middlebrook 7H9 broth (Gibco) supplemented with 10% (v/v) oleic acid-albumin-dextrose-catalase (OADC; Difco), 0.5% (v/v) glycerol and 0.05% (v/v) Tween 80 (Sigma-Aldrich) and stored in 1 ml aliquots at −80 °C.

### In vitro PAR formation
Aliquots of N-methyl-N'-nitro-N-nitrosoguanidine (MNNG; N-12560, Chem Service Inc) were stored at −20 °C. Human peripheral mononuclear cells (PBMCs) from two healthy donors were generously provided by Dr. Andrea Cox (Johns Hopkins, Baltimore). THP-1 (ATCC® TIB-202), Raw 264.7 (ATCC® TIB-71), HeLa (ATCC® CCL-2) and J774A.1 (ATCC® TIB-67) were purchased from ATCC. Cells were cultured in RPMI 1640 with GlutaMAX (Gibco) supplemented with 10% heat-inactivated fetal bovine serum of human origin (FBS; Gibco). THP-1 cells were differentiated into adherent macrophage-like cells by

overnight incubation in RPMI Glutamax supplemented with 10% FBS and 50 ng/ml PMA.

For MNNG-, TNF- or LPS-stimulated PAR formation, cells were seeded in 12-well plates ($5 \times 10^5$ cells/well in 2 ml) overnight, washed once, and exposed to 50 μM MNNG, 75 ng/ml TNFα or 500 ng/ml LPS in RPMI for 10, 30 or 60 minutes to activate PARP1. At the end of treatment, cells were washed once with ice-cold PBS and incubated in 75 μl denaturing sample buffer (69.45 mM Tris-HCl pH 6.8, 11.1% (v/v) glycerol, 1.1% (v/v) LDS, 0.005% (v/v) bromophenol blue, 2.5% (v/v) ß-mercaptoethanol) for one min on ice before cells were pipet-lysed, lifted and denatured at 95 °C for 10 min. Lysates were chilled on ice and stored at −80 °C for immunoblot analysis.

For *M.tb*-induced PAR formation, THP1 cells ($1 \times 10^6$ cells/well) were differentiated in 6-well plates in 2 ml RPMI 10% FBS supplemented with 50 ng/ml PMA overnight, washed once and incubated in RPMI 10% FBS without PMA for another night before infecting the following day. Differentiated THP1 cells were infected by adding $5 \times 10^6$ (MOI 5) or $1 \times 10^7$ (MOI 10) *M.tb* H37Rv, H37Ra, *M. smegmatis* or *M. bovis* BCG bacilli in OADC-supplemented 7H9 broth to corresponding wells. After 4 hours (time 0), cells were washed twice with PBS to remove any remaining extracellular bacteria, and infected cells were incubated in RPMI 10% FBS for 30 min (0.5 hpi) to 48 hours (48 hpi) before harvesting cells. At the end of treatment, cells were washed once with ice-cold PBS and incubated in 125 μl denaturing sample buffer for 1 min on ice before cells were pipet-lysed, lifted, and denatured at 95 °C for 30 min (to heat-kill *M.tb*). Lysates were chilled on ice and stored at −80 °C for immunoblot analysis.

### Animals
C3HeB/FeJ (stock #658) mice were purchased from The Jackson Laboratory. PARP1-deficient 129S-*Parp1tm1Zqw*/J mice (stock #002779, Jackson Labs) were bred in-house. PARP1 disruption was routinely validated by PCR as described by the supplier (protocol 22839, version 2.3) using the common forward primer (5′-CATGTTCGATGGGAAAGTCCC-3′), a WT reverse primer (5′-CCAGCGCAGCTCAGAGAAGCCA-3′) and a mutant reverse primer (5′-AGGTGAGATGACAGGAGATC-3′). Age-matched recommended control 129S1/SvlmJ mice (stock #002448, Jackson Labs) were purchased for each experiment. All animal procedures were approved by the Institutional Animal Care and Use Committee of the Johns Hopkins University School of Medicine.

### Aerosol infections
Mice were infected between 8 and 12 weeks of age via the aerosol route using the Glas-Col Inhalation Exposure System (Terre Haute, IN). A fresh aliquot of *M.tb* was used for each infection and diluted in sterile phosphate-buffered saline (PBS, pH 7.4) or 7H9 broth with OADC, glycerol, and Tween 80 at empirically determined factors to achieve the desired inoculum. To reduce intergroup variability, mice from all comparative groups were infected together or evenly distributed between infection cycles and randomly assigned to experimental arms. On the day after infection, 3–5 mice per cycle were sacrificed to determine the number of CFUs implanted into the lungs. The general appearance and body weight of mice were monitored at least weekly throughout all experiments. All infections, housing of infected mice, and handling of infectious materials were carried out under biosafety level 3 containment in dedicated facilities.

### Mouse treatments
Talazoparib (BMN-673; CAS no. 1207456-01-6, Medchemexpress LLC) was reconstituted in HPLC-grade DMSO (Sigma-Aldrich) and stored in 10 mM aliquots at −20 °C. DMSO aliquots were frozen at the same time to prepare vehicle control solutions. For PARP inhibition in mice, a talazoparib solution was prepared fresh daily in 0.5% low-viscosity carboxymethyl cellulose (CMC; Sigma-Aldrich). Solutions of pyrazinamide (PZA) and rifampin (RIF) in distilled

water were prepared weekly and stored at 4 °C. The PZA solution was gently heated in a 55 °C water bath and vortexed to dissolve prior to treating mice. All drugs were administered as indicated once daily by orogastric gavage, in a total volume of 0.2 ml per treatment. To minimize drug-drug interactions, RIF was administered at least 1 h before PZA.

## Tissue collection and bacterial enumeration

Mice were sacrificed at predetermined intervals and the total body weight was recorded. Lungs were aseptically removed and sectioned for bacterial enumeration (A; right lung lobes) and PAR, cytokine, and chemokine quantification (B; left lung lobes). The weights of the intact lung and the (B) section were recorded and used to estimate the total bacterial burden as described below. For bacterial enumeration, lungs (A) were placed in 2.5 ml sterile PBS for 24–48 h at 4 °C, examined for gross pathology, and manually homogenized. Homogenates were serial-diluted, and 0.5 ml plated on Middlebrook 7H11 agar (Difco) supplemented with 10% (v/v) OADC, 0.5% (v/v) glycerol, 10 mg/ml cycloheximide, 50 mg/ml carbenicillin, 25 mg/ml polymyxin B and 20 mg/ml trimethoprim (Sigma-Aldrich). Plates were incubated at 37 °C for 3–4 weeks before colonies were counted. Colony numbers were adjusted by the plating, dilution, and dissection factors and log-transformed to estimate the total colony-forming units (CFUs) per organ (CFUs = $\log_{10}$ [(number of colonies)*(5)*(A + B)/(A)] + [dilution factor]).

## Immunoblot and PAR analysis

PARP1 activity was assessed by comparing poly(ADP-ribose) (PAR) levels in experimental and control samples by immunoblot[48,49]. For PAR detection in cell lines, cells were washed once with ice-cold PBS and incubated in 75 μl denaturing sample buffer (69.45 mM Tris-HCl pH 6.8, 11.1% (v/v) glycerol, 1.1% (v/v) LDS, 0.005% (v/v) bromophenol blue, 2.5% (v/v) ß-mercaptoethanol) for one min on ice before cells were pipet-lysed, lifted and denatured at 95 °C for 10 min. For PAR detection in mouse tissues, lung sections were placed in chilled extraction buffer (50 mM glucose, 25 mM Tris-HCl pH 8.0, 10 mM EDTA) with protease inhibitor cocktail (Sigma-Aldrich) and 1.0 mm zirconia beads (5.5 g/cc; Biospec Products) and immediately homogenized by bead-beating in a mini-beadbeater (Biospec Products) in 30 s intervals at 4,800 RPM. Homogenates were placed on ice for 10 min, mixed with denaturing sample buffer (69.45 mM Tris-HCl pH 6.8, 11.1% (v/v) glycerol, 1.1% (v/v) LDS, 0.005% (v/v) bromophenol blue, 2.5% (v/v) ß-mercaptoethanol) and denatured at 90 °C for 15 min. Lysates were chilled on ice and stored at −80 °C for immunoblot analysis.

Protein concentrations were determined by CB-X protein assay (G-Biosciences) according to the manufacturer's instructions. Cell lysates or lung homogenates in denaturing sample buffer (10–20 μg total protein) were separated by SDS-PAGE on 4–15% Mini Protean TGX gels (Bio-Rad) and transferred electrophoretically to a 0.45 μm nitrocellulose blotting membrane (GE Healthcare) at 100 V for 75 min at 4 °C. To assess transfer efficiency, the membrane was immersed in 0.1% (w/v) Ponceau S in 5% (v/v) acetic acid for 5 min, rinsed with distilled water, and imaged. The membrane was de-stained in 0.1 M sodium hydroxide, rinsed with running distilled water for 2–3 min, and washed with tris-buffered saline containing 0.1% (v/v) Tween 20 (TBST). The membrane was then blocked in TBST containing 5% (w/v) nonfat dry milk for 1 h at room temperature (RT), washed in TBST and incubated with a recombinant human monoclonal anti-PAR antibody (1:2500 in 5% nonfat dry milk-TBST; clones #19 and #21, highly specific for target-bound PAR, custom-designed at Bio-Rad AbD Serotec GmbH)[48] with gentle agitation overnight at 4 °C. Following primary antibody incubation, the membrane was washed, incubated with HRP-conjugated goat anti-human IgG (Fab')2 (1:5000 in 1% nonfat dry milk-TBST; Abcam) for 1 h at RT and visualized with KwikQuant Ultra digital

ECL substrate using the KwikQuant digital imager (Kindle Biosciences, LLC).

After visualizing PAR bands, the membrane was washed with TBS and stripped in a 200 mM glycine solution (pH 2.5) for 1 h at RT. The membrane was washed with TBS and TBST, blocked in TBST containing 5% (w/v) nonfat dry milk for 1 h at RT, washed with TBST, incubated with HRP-conjugated mouse monoclonal anti-beta Actin antibody [AC-15] (1:50,000 in 5% (w/v) BSA in TBST; Abcam) for 0.5–1 h at RT and visualized as described for PAR detection. Digital images were converted to black-and-white using Photoshop, and relative band intensities were quantified using ImageJ (version 1.52a) as described[82]. PARP1 activity was defined as the intensity of high-molecular-weight (72–250 kDa) PAR bands after normalizing to β-actin and was expressed as fold change relative to uninfected or untreated control samples on the same immunoblot. Uncropped and unprocessed blots are supplied in the source data file (main display items) or at the end of the Supplementary Data file (Supplementary Figs.).

## Cytokine and chemokine analysis

Lung cytokines/chemokines (IFNγ, IL-1ß, TNFα, IL-6, IL-10, IL-12(p70), MCP-1/CCL2, RANTES/CCL5, LPS-induced CXC chemokine (LIX), KC/CXCL1) were analyzed by Luminex multiplex bead assay on a Bio-Plex 200 platform (Bio-Rad) with a mouse cytokine/chemokine magnetic bead panel (MCYTOMAG-70K, lot # 3737812; Millipore) according to the manufacturer's instructions. Lung tissue was disaggregated in a mini-beadbeater (Biospec Products) at 4800 RPM with 1.0 mm zirconia beads (5.5 g/cc; Biospec Products) in sterile PBS, incubated on ice for 5 min and centrifuged at 12,000×g for 10 min. Supernatants were filter-sterilized through 0.22 μm cellulose acetate Spin-X centrifuge tube filters (Costar) and stored at −80 °C. Protein concentrations were determined by Quick Start™ Bradford protein assay (BioRad) according to the manufacturer's instructions, and samples were diluted to 1 mg/ml for analysis. Results are presented as concentrations (pg/ml) or fold change relative to uninfected lungs from the same group.

## RNA extraction and quantitative real-time PCR (qPCR)

Total RNA was isolated from flash-frozen *M.tb*-infected mouse lung tissue using the RNeasy Mini kit (Qiagen) and contaminating DNA was removed by digestion with DNase I (Promega) according to the manufacturer's instructions. cDNA was synthesized using the High-Capacity cDNA Reverse Transcription Kit (Thermo-Fisher Scientific). Quantitative real-time PCR was performed using Real-time PCR Fast SYBR Green Master Mix (Thermo-Fisher Scientific) on a StepOne Plus Real-time PCR instrument (Applied Biosystems). Primers (Supplementary Table 3) were designed using the GenScript Real-time PCR Primer Design tool (TaqMan) and synthesized by Integrated DNA Technologies. Data were analyzed using StepOne Software v2.3 (Applied Biosystems). Cycle threshold values were normalized to a housekeeping gene (*Gapdh*) by subtracting the corresponding *Gapdh* CT from the gene of interest CT for all samples (ΔCT). ΔΔCT was obtained by subtracting the average vehicle ΔCT from the ΔCT of each sample. Transcript fold change presented in the results is equal to $2^{-\Delta\Delta CT}$ relative to vehicle-treated mice.

## Histopathology and inflammation analysis

For histology, intact lungs were fixed by immersion in 10% neutral-buffered formalin for 48 h, paraffin-embedded, sectioned, and stained with hematoxylin and eosin (H&E). Slides were digitally scanned at 40× on an Aperio AT turbo scanner console version 102.0.7.5 (Leica Biosystems). Image files were transferred using Concentric for Research version 2.2.4 (Proscia Inc). Digital slides were blinded to experimental grouping, randomized and analyzed using Aerie ImageScope software (Leica Biosystems). Region of interest (ROI) selection was performed manually by a boarded veterinary pathologist to outline areas of inflammation, characterized by increases in cellular density, loss or

distortion of normal lung architecture, obliteration of air spaces, and regional tinctorial changes. ROIs were then summed for each slide, and divided by the total lung area per slide to calculate the ratio of inflamed to unaffected lung tissue.

## Immunohistochemistry (IHC) analysis

Consecutive sections of formalin-fixed, paraffin-embedded lungs were immunostained with validated antibodies against mouse F4/80 (macrophages), Ly6G/6 C (Gr-1; neutrophils), vimentin (fibroblasts), CD4 or CD8 (CD4+/CD8+ T-cells, respectively) by the Oncology Tissue Services core facility (Johns Hopkins, Baltimore, MD). Quantitative analysis of IHC markers (vimentin, CD4, CD8, GR-1, and F4-80) was performed using QuPath digital pathology image analysis software[83] to calculate the percent immuno-positive area of each marker in a cross-section of the whole lung (% positive area/lung), as well as to compare the percent positive area of each inflammatory marker in foci of lung consolidation (% positive area/inflammation). Briefly, the total lung area of each slide was calculated by setting a threshold value for tissue detection that gated out the background; automated tissue detection of the ROI was confirmed by manual visual assessment (Channel: Average channels, smoothing sigma 3, Threshold 225). Similarly, to calculate the immuno-positive area within the lung ROI, a threshold was set that enabled the detection of the DAB chromophore, and ignored staining below this threshold (Channel: DAB, Smoothing sigma 1, Threshold 0.14). The same thresholds and detection settings were used for each IHC marker in all slides.

For calculating the percent positive area in foci of consolidation, three independent foci of parenchymal pulmonary consolidation >500 μm in diameter were manually selected per slide at 4× magnification. Areas with extensive central necrosis were avoided due to excessive background staining. Using serial sections of lung tissue, the same approximate foci were evaluated for each IHC marker. A fixed, 500 μm diameter spherical ROI was manually centered around these inflammatory foci and the DAB-positive area was calculated using the aforementioned threshold settings. In lungs that lacked lesions >500 μm in diameter, the region of interest was centered on the largest areas of consolidation present in each lung. The DAB-positive area of the three 500 μm diameter foci was averaged to generate the % positive area of each marker in regions of pulmonary consolidation.

## Statistical analyses

Statistical analyses were performed using Prism version 9.2.0 for Windows (GraphPad). Statistical tests used are indicated in the figure legends. Differences between two group means were assessed by unpaired, two-tailed $t$-test. Changes in cellular PAR levels were compared by repeated measures ANOVA with Dunnett's multiple comparisons test. Data sets containing three or more groups were analyzed by one- or two-way analysis of variance (ANOVA) with Tukey's, Dunnett's, or Šidák's multiple comparisons test or uncorrected Fisher's Least Significant Difference (LSD) test where indicated. Normality was assessed by the D'Agostino & Pearson omnibus normality test. Data that did not follow a normal (Gaussian) distribution were analyzed by the Kruskal–Wallis nonparametric test with uncorrected Dunn's multiple comparisons test and are plotted as median ± interquartile range, as indicated in the figure legends. CFU counts were $\log_{10}$-transformed prior to analysis. A $p$ value below 0.05 was considered significant. Data represent mean ± SEM. Illustrations were generated using the open-source vector graphics editor Inkscape for Windows (v. 0.92.4).

## Reporting summary

Further information on research design is available in the Nature Portfolio Reporting Summary linked to this article.

## Data availability

Source data generated in this study are provided with this paper in the Source Data file. A repository of the digitally scanned, H&E-stained lung sections used for histology analysis is publicly available at https://tinyurl.com/4az6ueks. A repository of the digitally scanned IHC-stained lung sections is publicly available at https://tinyurl.com/3v384esc, and the decoded slide IDs are listed in the Source Data file. Uncropped and unprocessed Western blots are provided. Source data are provided in this paper.

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

## Acknowledgements

This work was supported by National Institute of Health (NIH) grants R21-AI137659, R21-AI130595, R01-HL133190 and R01-AI037856 (W.R.B.), F31-HL140812 (SK), P50-NS038377 (V.L.D.), RF1-AG059686 (T.M.D. and V.L.D.) and funding from the JPB foundation (T.M.D.). T.M.D. is the Leonard and Madlyn Abramson Professor in Neurodegenerative Diseases. We thank Eric Nuermberger and Rokeya Tasneen for generously sharing the PZA-resistant *pncA* mutant *M.tb* H37RvΔ*pncA* (A146V), and Elizabeth Ihms, DVM, PhD, DACVP (Dane & Dutchy LLC) and Chris Thoburn (SKCCC Immune Monitoring Core) for helpful discussions and assistance with histopathology and for cytokine/chemokine analysis, respectively. We are especially grateful to Nicole Ammerman, Deepak Almeida, Sandeep Tyagi, Rokeya Tasneen, Si-Yang Li, Pankaj Prasad, Akshay Rohilla, Monika Looney, Will Matern, Michael Urbanowski, Cynthia Korin Bullen, Alok Kumar Singh, Shichun Lun, Kristina Bigelow and Shiqi Xiao for their assistance with the mouse experiments presented in this paper.

## Author contributions

W.R.B., V.L.D and S.K. conceived the project. W.R.B., S.K., V.L.D., B.K. and P.K. designed the experiments. T.M.D., V.L.D and B.K. provided critical reagents and technical expertise. S.K., M.G., P.K., L.F. and E.A.I. carried out experiments and analyzed data. W.R.B, G.S. and S.K. interpreted the data, prepared the tables and figures, and wrote the manuscript. All authors read, provided feedback on, and approved the paper.

## Competing interests

The authors declare no competing interests.
