## [Peer Review File · Nature Communications]

Inhibition of host PARP1 contributes to the anti-inflammatory and antitubercular activity of pyrazinamideEditorial Note: Parts of this Peer Review File have been redacted as indicated to remove third-party material where no permission to publish could be obtained.

REVIEWER COMMENTS

Reviewer #1 (Remarks to the Author):

Shortening TB therapies is a priority and host-directed therapies have increasing relevance in this context. PZA reduces therapy length, displays modest bactericidal activity and is an anti-inflammatory molecule. Thus, PZA appears as a HDT of interest, already in use, and which mechanism may lead to important progress in the area. This study sheds light on the mechanism underlying PZA activity in TB, by identifying PARP1 as its host target, and demonstrates the potential of modulating inflammation for better outcomes. As such, this study provides novel data and is of interest to the field; it may also offer novel clues on how to design better, shorter HDT to TB. There are however some questions that deserve attention and could improve the manuscript if answered, namely by making a stronger link between the sterilizing effect of PZA and the regulation of inflammation/pathology.

The authors predict binding of PZA to PARP1 and show that indeed PZA inhibits PARP1 activity in THP1 cells. The quality of the western blot images shown in Fig 2b is not great. Considering the excellent quality of those in Fig 3, consider to improve these (Minor point). Data presented place Tp as the best inhibitor of PARP1.

The authors then move to in vivo experiments. They show that PARP1 activity is increased in infected mice (Supp Fig 2). This increase is in fact quite variable among animals. Does this correlate with mouse weight, bacterial burden, lung pathology? In a similar note, both PZA and Tp treatment of infected mice contributed to decreased activities of PARP1, with a fairly large dispersion in the levels of inhibition, particularly in the case of PZA (Fig 3c). Then, when analysing CFU data, in the case of PZA, two groups are seen in the graph (Fig 3d). What is (if any) the correlation between better/worse bacterial control and PARP1 inhibition? As shown for CFU (Fig 3d), it would be informative to have in the graph of Fig 3c the variation in PAR levels of the vehicle group represented (Minor point).

An interesting finding of this set of experiments is that although both PZA and Tp seem to inhibit PARP1, their effect on bacterial burden is almost opposite. Furthermore, combination of Tp and RIF led to a significant increase in CFU as compared to RIF alone (Fig 4b). This aspect should be discussed, as it raises concerns on a possible clinical application of Tp, as the authors suggest in their discussion.

Data in Fig 4 show that treatment with RIF+Tp or RIF+PZA visibly reduce the area of inflammation and % lung involvement in infected mice. It would be interesting to investigate what type of inflammatory mechanisms are being targeted, and whether they are the same in both cases (the fact that lesion number and lesion size vary differently in PZA vs Tp therapy may suggest different mechanisms of action, something that should be at least discussed). Deregulated neutrophil and type I IFN responses were shown to underlie the susceptibility of C3HeB/FeJ to *M. tuberculosis* infection. These are prime candidates to measure. The authors present levels of IFN- γ and IL-1b as correlates of inflammation and candidate molecules to be modulated by PARP1 inhibition. However, both molecules are already decreased in RIF only mice as compared to untreated (Supp Fig 4c), which may suggest that the main driver for the decreased production is the control of the bacterial burden.

In the last set of experiments, the authors use WT and PARP1 deficient mice. In Fig 5 the authors show that PZA treatment, independently of CFU control, generally decreases cytokine/chemokine responses in WT mice, but fails to do so in PARP1^{-/-}. Data in Supp Fig 5 (direct comparison between groups) are not as convincing as those presented in Fig 5d (based on fold change). It would be important to support this anti-inflammatory activity of PZA through PARP1 in complementary ways, namely, by presenting histologic analysis and quantification of the lungs, as shown in the previous set of experiments. Similarly, in Fig 6 only bacterial burdens are shown, but given the links presented in the rest of the study, it would be important to address the immune response and lung pathology in this case.

Reviewer #2 (Remarks to the Author):

In this study, the authors suggest that inhibition of the inflammatory cellular regulator, PARP1, could support and enhance the *in vivo* effect of pyrazinamide (PZA) in *M. tuberculosis* (Mtb) infected mice by reducing inflammation. The work is divided into an *in vitro* part using recombinant PARP1 or monocytic cell lines, and an *in vivo* part involving Mtb infection of C3H mice and treatment with the first-line anti-TB antibiotics, PZA and rifampicin (RIF), in the presence or absence of the PARP1/2 inhibitor, talazoparib (Tp). The *in vivo* work also uses relevant Mtb mutant strains (PZA-resistant) and PARP1 deficient mice to investigate the single- and combined effects of Tp and PZA and/or RIF. The authors conclude that PZA, similar to commercial PARP1 inhibitors, binds to the active site of PARP1. PZA further inhibits PARP1 activity in THP-1 cells and in Mtb-infected mouse lungs after 2-months *in vivo* administration. Treatment with Tp alone or in combination with RIF, reduces lung inflammation and pathology in Mtb-infected mice, even though bacterial loads in the lungs are relatively higher compared to other treatment groups. Luminex multiplex of lung homogenates shows that Tp treatment results in down-regulation of IL-1b and IL-12, while a PARP1-dependent effect on a number of other cytokines and chemokines involved in inflammation/Th1 immunity is also proposed (non-significant observations).

Overall, the hypothesis that inhibition of a protein involved in inflammatory processes could be targeted for potential host-directed therapy in TB is interesting and highly relevant in a time of increasing antibiotic resistance. I also appreciate these difficult and time-consuming experiments with virulent Mtb, which are very important to obtain new knowledge of how immune responses in TB can be modulated. However, I'm not convinced by the data itself (please, see my specific comments below) and would like to see additional experiments to support the author's conclusions. Potential limitations of the study and results should be included in the Discussion. Moreover, I consider the discussion around proinflammatory cytokines in TB to be a bit insufficient and too general. While the manuscript is well-written and easy to read and follow (plus for schematic illustrations of experimental set up in each figure), the findings should be balanced with an introduction and discussion around proinflammatory vs Th1 immunity and anti-inflammatory responses in TB, as inflammation is also a necessary requirement to prime both macrophages and specific T cell responses that could enhance intracellular eradication of Mtb.

Specific comments:

1. I have several comment and questions with regards to the *in vitro* experiments in Figure 2, using monocytic cell lines and the PARP1 activator, MNNG.
 - o How come the authors didn't compare uninfected to Mtb-infected macrophages to confirm that PARP activity is upregulated in infected cells and that PZA, similar to Tp and NAM, could down-regulate PAR formation? This would be the logical experiment before studies of *in vivo* Mtb-infected mouse lungs.
 - o In addition, I believe these data would be significantly stronger if monocyte-derived macrophages were used instead of cell lines. There are standard protocols for this, and the authors have access to PBMCs from healthy donor blood, so this should be feasible.
 - o Viewing the immunoblots of PAR in Fig. 2b, it is not clear to me where specific band(s) are located? The lanes look more like smears rather than showing specific band as for example in Fig. 3b. Is it possible to visualize the molecular weight control as well, for clarity? At least, the authors should discuss the difference in the bands obtained with cell lines *in vitro* and with lung homogenates *in vivo*.
2. Figure 3b illustrates that there is obviously a great variability in PAR formation comparing Mtb-infected individuals ie. n=5 mice/group. Is that expected and something that has been shown in other studies/diseases?
3. In Figure 4b, why wasn't a group with Tp+PZA included? Perhaps I missed out on something, but it is quite obvious that this group should have been included in the CFU graphs similar to Tp+RIF. Even if the difference in CFU counts between RIF alone and Tp+RIF is not significant, it is quite clear that the latter group has relatively higher CFU counts. I'd appreciate an extended discussion on this phenomenon.

4. Inflammation and lung involvement presented in Figure 4d-e, are two important criteria used to illustrate that inhibition of PARP1 activity results in reduced pulmonary pathology in TB disease. In the Materials and methods, it should be explained in more detail how these parameters were quantified. Here, I would recommend to use a validated scoring or grading system to authenticate the data. In addition, only H&E stains were used to study inflammation and lung involvement, while immunostainings for eg. neutrophils (elastase), CD4 and CD8 T cells, and myeloid cells (inflammatory and anti-inflammatory monocytes and macrophages), would significantly strengthen these data. The authors have access to formalin-fixed lung tissue so this should be feasible. An interesting option that is perhaps out of the scope of this manuscript, is also to use lung homogenates for flowcytometry, to obtain a systematic view on the immune cell subsets present in the lungs after respective treatment. These analyses would enable a more detailed assessment of immunopathology in Mtb-infected mice treated with PZA, Tp etc.

5. Regarding Figure 5c-d, and as explained in the summary, I'd like to obtain a nuanced description of the role of ie. IL-1b (activation of neutrophils) compared to IL-12 (activation of DCs), TNFa (activation of macrophages) or IFN-g (activation of macrophages and T cells). As far as I understand it, there is a trend towards down-regulated levels also of IL-10 in WT and PARP1-/- mice (Supp. Fig. 5). Are any cytokines/chemokines or other mediators up-regulated by PARP1 inhibition? What is the definition of pathological (pro)inflammation in TB and when or how is it important to block inflammation? Assessment of other inflammatory mediators known to be important in Mtb-infected cells, such as iNOS/NO, ROS, or autophagy, would also add to an understanding of the function of PZA in PARP1-dependent regulation of TB immunity.

6. The illustration in Figure 6e is nice, but adjunctive PARP1 inhibition with ie. Tp+RIF, would then result in intermediate/relatively higher Mtb load but reduced lung damage, perhaps this should be included somehow.

7. My final comment is that it that it would have been good to obtain viability data from the mice used in the respective groups, especially given the slightly higher bacterial loads in Tp+RIF compared to RIF alone. I realize this would require a separate experiment, but perhaps this could be done for the most interesting groups/drug combinations. Even if PARP1 inhibition can reduce immunopathology in TB, it is still not clear to me how PARP inhibitors could be used as adjunct treatment in MDR-TB patients, in the absence of PZA (as mentioned in the Discussion).

8. Statistical comment: Please, note that data not passing a normality test should be presented (median +/- IQR) and analysed using non-parametric methods.

Reviewer #3 (Remarks to the Author):

Inhibition of host PARP1 contributes to the anti-inflammatory and antitubercular activity of pyrazinamide

Krug et al. identify PARP1 as a host factor that could be targeted by the anti-TB compound pyrazinamide (PZA), at least partly explaining the effects of PZA use. The authors indicate that there is literature supporting PZA acting against a host target to influence the inflammation response during treatment, but that the target has not been identified. The study first investigates whether PZA will bind to the PARP1 active site, and they test whether PZA lowers poly(ADP-ribose) production in cells. Using mouse models infected with TB, the study tests the influence of PZA (alone or in combination with other compounds) on poly(ADP-ribose) production, bacterial burden in lungs, lung histopathology, and immune response. The overall story is compelling; however, the presentation of the data and limitations could be improved.

Here are some specific comments:

Figure 1c. What do the fluorescence v. temperature plots look like (rather than the derivative of the fluorescence signal)? The derivative plots typically will have a stronger single peak. It seems

these experiments were performed with the entire PARP1 molecule (not just the catalytic domain), and this could perhaps explain the appearance of the data, since full-length PARP1 will have several domains unfolding and not necessarily at the same temperature. The errors associated with the T_m measurements should be shown, and the plot of the raw data would be helpful. It would also be helpful to perform an in vitro inhibition assay with these compounds, rather than moving directly to cells.

page 6, the description of Supplementary Figure 2 requires some nuance, since the PAR levels were quite variable in infected lung cells. It seems that there is no statistical difference, but the increase is referred to as robust.

"By structural alignment, we found that PZA is predicted to bind the PARP1 ART fold in the same manner as BAD, and that it even forms one additional bond (Phe897) (Figure 1 b, right)"

I would tweak this sentence in the following way to be more accurate:

"... and that it is predicted to form an additional bond (Phe897) based on our modeling..." The model seems reasonable, but even small changes can lead to massive rearrangements in how molecules bind to active sites, so it is important to stress that a model is being presented.

Figure 3, there needs to be some discussion of the variability observed within treatment groups. What can explain the Tp-treated mouse with substantial PAR levels, for example? The results do not seem to be as clear cut as the presentation in the text. For the statistical analysis, it would be useful to list all of the P values for the comparisons made.

page 6, "and no discernable inhibitory effects at eight times the concentration used to treat mice (Supplementary Table 1)"

Is this data actually shown? I could not find it.

Figure 5, why not just show the P values across all samples?

Figure 5 data. These mouse experiments were done using TB with resistance to PZA in order to focus on the host effects of PZA. If the only host effect was to target PARP1, it seems that PARP1 knockout should mimic PZA treatment of WT mice. However, the PARP1 knockout mice appear to be quite different from PZA-treated WT mice. There might be a good explanation for this, but I found this to be confusing and counter intuitive.

Along the same lines, can it be assumed that the PARP1-directed effects of PZA are captured in Figure 6d, by comparing PZA-treated WT to PZA-treated PARP1^{-/-} ? I was left without a strong feeling for how much the PARP1-directed effects of PZA contribute. Perhaps this can be elaborated on.

What is the basis for the term "sterilizing ability"? Does this refer to antibacterial activity? Worth a short sentence for those not familiar with this term.

Re: **NCOMMS-22-0952** – Inhibition of host PARP1 contributes to the anti-inflammatory and antitubercular activity of pyrazinamide

We would like to thank the reviewers for their helpful and insightful comments. We have now revised the manuscript extensively attending to each point, and added or expanded 20 new figure panels and 2 new data tables. In particular, we have characterized immune responses and immune cell populations to reveal additional mechanistic insight into PARP1 activity and inhibition during TB therapy. We feel that this revised paper is now substantially improved.

Reviewer 1 (Remarks to the Author):

Shortening TB therapies is a priority and host-directed therapies have increasing relevance in this context. PZA reduces therapy length, displays modest bactericidal activity and is an anti-inflammatory molecule. Thus, PZA appears as a HDT of interest, already in use, and which mechanism may lead to important progress in the area. This study sheds light on the mechanism underlying PZA activity in TB, by identifying PARP1 as its host target, and demonstrates the potential of modulating inflammation for better outcomes. As such, this study provides novel data and is of interest to the field; it may also offer novel clues on how to design better, shorter HDT to TB. There are however some questions that deserve attention and could improve the manuscript if answered, namely by making a stronger link between the sterilizing effect of PZA and the regulation of inflammation/pathology.

- 1. The authors predict binding of PZA to PARP1 and show that indeed PZA inhibits PARP1 activity in THP1 cells. The quality of the western blot images shown in Fig 2b is not great. Considering the excellent quality of those in Fig 3, consider to improve these (Minor point). Data presented place Tp as the best inhibitor of PARP1.*

We thank the reviewer for this comment. Similar to a phospho-blot, PAR immunoblots are a smear rather than the defined band you would expect from a protein Western blot. The PAR smear that is observed in cultured cells depicted in Figure 2b (and Supplementary Figure 2) represents a wide range of PARP1-generated ADP-ribose polymers that vary greatly in length. These PAR blots are consistent with those reported in other studies- with increased polymerization correlating with the strength of activation (e.g., Brunyanszki et al., Mol Pharmacol (2014); Andrabi et al, PNAS (2014); see images below).

In contrast, we detected a much narrower range of PAR polymers in lungs (Figure 3b and Supplementary Figure 3a) that were most pronounced around the site of PARP1-auto-PARylation, with only a faint smear above and below. We speculate that this may reflect differences in the strength of PARP1 activation during infection as opposed to following stimulation with the potent PARP1 activator MNNG; an overall weaker PAR signal in a heterogenous tissue sample compared with a homogenous population of macrophages in a dish; or the degradation of ADP-ribose polymers by enzymes and proteases during tissue processing. We have elaborated on this in the text (lines 127-129, and 393-397). We agree that Tp is a more potent inhibitor of PARP1 than PZA. We mention so in lines 141-142.

[REDACTED]

[REDACTED]

Andrabi et al., PNAS (2014)

Brunyanszki et al., Mol Pharmacol (2014)

2. *The authors then move to in vivo experiments. They show that PARP1 activity is increased in infected mice (Supp Fig 2). This increase is in fact quite variable among animals. Does this correlate with mouse weight, bacterial burden, lung pathology?*

We thank the reviewer for this observation. Further analyses showed that PAR levels 1 month post infection in fact correlated with bacterial burden, but not with body, lung or spleen weights. We have now indicated this in lines 149-152 in the text. We also now include the regression plots as new Supplementary Figures 3c and 3d.

3. *In a similar note, both PZA and Tp treatment of infected mice contributed to decreased activities of PARP1, with a fairly large dispersion in the levels of inhibition, particularly in the case of PZA (Fig 3c). Then, when analysing CFU data, in the case of PZA, two groups are seen in the graph (Fig 3d). What is (if any) the correlation between better/worse bacterial control and PARP1 inhibition?*

We thank the reviewer for these astute observations. It is well known that while PZA is uniformly bactericidal in BALB/c and C57Bl/6 mice, PZA-treated C3HeB/FeJ mice form two distinct groups, with the majority of mice displaying CFU reductions comparable to BALB/c or C57Bl/6 mice while a subset appears to be PZA-nonresponsive (much like athymic nude mice) for reasons that are not fully understood (reviewed by Lamont and Baughn, 2019, DOI: 10.1016/j.ebiom.2019.10.014). This “selective inactivity” of PZA in C3HeB/FeJ mice is not explained by bacterial drug resistance but by the host environment and/or its impact on bacterial dynamics (Lanoix et al., 2016, DOI: 10.1128/aac.01370-15; Blanc et al., 2018, DOI: 10.1084/jem.20180518). The host environment could affect the bactericidal activity of PZA directly (e.g., lesion pH microgradients affecting PZA efficacy; inactivity against extra- vs. intracellular bacilli) or indirectly (e.g., heterogenous environments or immune pressures creating metabolically distinct bacterial subpopulations with differing phenotypic PZA susceptibilities). Since PZA is ineffective in immunocompromised mice, another intriguing explanation is that PZA’s *modulation* of the host environment affects this dichotomy (e.g., by enhancing immune-mediated bacterial clearance). Here, we thus chose the C3HeB/FeJ mouse model of TB, which more closely recapitulates the lesion diversity of human TB than BALB/c or C57Bl/6 mice, to investigate whether PARP inhibition contributes to the bimodal PZA responsiveness.

As recommended, we have now analyzed the relationship between PAR levels and bacterial burden by treatment group and added multiple panels of regression plots in Supplementary Figure 4 c-e. These analyses proved extremely insightful and revealed a similarly bimodal impact on PARP1 activity (inhibited in 6/8 mice) as on CFU (reduced in 5/8 mice). PZA reduced

both PAR and CFU in 4/8 (50%), PAR only in 2/8 (25%) and CFU only in 1/8 (12.5%) mice, while affecting neither in 1/8 (12.5%) mice. In fact, 80% (4/5) of the mice in which PZA reduced bacterial burdens also had potentially reduced PAR levels, indicating that PZA efficacy most often coincides with PARP1 inhibition. In contrast, bacterial killing by RIF was independent of PARP1 activity, and PARP inhibition without effective antibiotic activity appeared to antagonize bacterial clearance. These findings suggest that PARP1 inhibition is not required for but may potentiate the bactericidal efficacy of PZA. We have elaborated on these findings in the text (lines 160 – 179 and 377-381).

4. *As shown for CFU (Fig 3d), it would be informative to have in the graph of Fig 3c the variation in PAR levels of the vehicle group represented (Minor point).*

We thank the reviewer for this suggestion. The nature of this assay makes it hard to compare between blots, so all comparisons are internally controlled (*i.e.*, we included a set of vehicle control samples on every blot). There was minimal variation in the vehicle control groups so we omitted them from their main figure for clarity but we have now added a new Supplementary Figure 4a (line 154 in the text) showing variations in PAR levels in vehicle groups.

5. *An interesting finding of this set of experiments is that although both PZA and Tp seem to inhibit PARP1, their effect on bacterial burden is almost opposite. Furthermore, combination of Tp and RIF led to a significant increase in CFU as compared to RIF alone (Fig 4b). This aspect should be discussed, as it raises concerns on a possible clinical application of Tp, as the authors suggest in their discussion.*

We thank the reviewer for this valid concern. Since Tp only has host-directed effects, and RIF primarily antimycobacterial effects, we attempted to emulate the dual host- and antimycobacterial effects of PZA with combined RIF + Tp (lines 189-192). Further analyses (new Supplementary Figure 4 b, d-e) confirm the reviewer's suspicion that PARP inhibition without adequate antibiotics appears to antagonize bacterial clearance, and we have addressed this in the text (lines 196-198). These analyses showed that in the absence of effective antibiotics (vehicle, Tp), PAR levels were trending toward an inverse correlation with CFU (new Supplementary Figure 4 e), suggesting that PARP1 may contribute to TB containment during the chronic phase of infection (lines 331-336). However, RIF + Tp-treated mice still had significantly reduced bacterial burdens compared to vehicle-treated mice, along with dramatically improved lung pathology compared to all other treatment groups (lines 208-210). We thus consider our study a proof-of-principle evaluation of adjunctive PARP inhibition in TB therapy and are optimistic that future studies can further optimize the Tp dosing strategy to minimize its negative impact on bacterial clearance while preserving its ability to accelerate the resolution of TB lung disease (discussed in lines 355-357).

6. *Data in Fig 4 show that treatment with RIF+Tp or RIF+PZA visibly reduce the area of inflammation and % lung involvement in infected mice. It would be interesting to investigate what type of inflammatory mechanisms are being targeted, and whether they are the same in both cases (the fact that lesion number and lesion size vary differently in PZA vs Tp therapy may suggest different mechanisms of action, something that should be at least discussed). Deregulated neutrophil and type I IFN responses were shown to underlie the susceptibility of C3HeB/FeJ to M. tuberculosis infection. These are prime candidates to measure. The authors present levels of IFN-g and IL-1b as correlates of inflammation and candidate molecules to be modulated by PARP1 inhibition. However, both molecules are*

already decreased in RIF only mice as compared to untreated (Supp Fig 4c), which may suggest that the main driver for the decreased production is the control of the bacterial burden.

We thank the reviewer for this valid concern. In response to this concern, besides IFN γ and IL-1 β , we also quantitated cell type composition (new Figure 4 f; new Supplementary Figures 6-7) and the expression of various immune mediators in the lungs of treated mice by ELISA and qPCR (new Supplementary Figure 8 b). We found that RIF+Tp or RIF+PZA treatment dramatically reduced neutrophil frequencies in the lung and granuloma (Supplementary Figure 6 d-e). Surprisingly, neutrophil frequencies were unaffected by RIF alone, even though RIF-treated mice had lower bacterial burdens than RIF+Tp-treated mice, or by Tp alone, indicating that 1) neutrophil frequencies were not reflective of bacterial burden and 2) adjunctive PARP inhibition can reduce neutrophil infiltration only when bacterial replication is controlled by antibiotics. In contrast, lung infiltration by fibroblasts, macrophages and CD8+ T-cells decreased in response to either bacterial killing or PARP inhibition, and most potently to both. In addition, PARP1 appears to be critical for CD4+ T-cell responses or expansion since Tp and RIF+Tp-treated mice had the overall lowest CD4 frequencies. Considering their protective role in TB, this observation offers a potential explanation for the impaired TB containment associated with PARP inhibition. Similarly, PZA or RIF treatment increased the expression of CXCR5 in lungs, a response that was attenuated by the addition of Tp (new Supplementary Figure 8 b). Since CD4+CXCR5+ T cells have been shown to promote protective immunity against tuberculosis (Slight et al., 2013, DOI: 10.1172/jci65728), we speculate that an overall reduction in protective CD4+ T-cell responses may be responsible for the increased bacterial burden in RIF+Tp-treated mice.

Although we saw no clear differences in TNF α protein levels in mice receiving *any* antibiotic, either PARP inhibition (Tp) or bacterial killing (RIF) modestly lowered TNF α expression at the transcriptional level but only the combination of both (PZA, RIF+Tp, RIF+PZA) resulted in significant downregulation (new Supplementary Figure 8 b). iNOS, but not MCP-1, expression followed a similar pattern. Unfortunately, IFN β levels were below the limit of quantification in our lung samples so we instead evaluated the expression of *Ifn β* and the type 1 IFN-inducible genes *Ifit1* and *Ifit3*, which are highly induced during *M.tb* infection and were recently identified as systemic biomarkers predictive of active TB (Qiu et al., 2022, DOI: 10.1186/s12931-022-02035-4). Remarkably, as shown in new Supplementary Figure 8 b, we found that PARP1 inhibition, especially when combined with antibiotics (PZA, RIF + Tp, or RIF + PZA), potently suppressed type I IFN gene expression (*Ifn β*) and signaling (*Ifit1*, *Ifit3*). In contrast, though RIF alone also dampened *Ifn β* expression it had no effect on the expression of type 1 IFN-inducible genes, indicating that PARP inhibition rather than a change in bacterial burden is the driver of these effects. Since type I IFNs are associated with TB susceptibility and pathogenesis, as pointed out by the reviewer, our findings implicate Type I IFN signaling as a primary target of adjunctive PARP1 inhibition and potential mechanism behind the dramatically improved lung pathology observed in RIF+Tp-treated mice.

The above findings, discussed in lines 215-226, 240-256, 336-339, and 344-359, suggest that some of the effects of PZA are mediated by its ability to reduce bacterial burden (reduced IL-1 β and MCP-1, increased CXCR5), while others are due to its ability to inhibit PARP-1 (reduced IFIT1 and IFIT3), or due a combination of both (reduced IFN γ , IFN β , TNF α and iNOS). The combination effect is best exemplified by comparing the effects of PZA alone with that of RIF+Tp vs. Tp or RIF alone (lines 189-192). We speculate that the divergent impact of Tp and PZA on lesion size/number is a reflection of immune modulation with (PZA) or without (Tp)

antibiotic control of infection, consistent with other observations in this study suggesting that PARP inhibition is independent of but complementary to the bactericidal effects of PZA.

7. *In the last set of experiments, the authors use WT and PARP1 deficient mice. In Fig 5 the authors show that PZA treatment, independently of CFU control, generally decreases cytokine/chemokine responses in WT mice, but fails to do so in PARP1^{-/-}. Data in Supp Fig 5 (direct comparison between groups) are not as convincing as those presented in Fig 5d (based on fold change). It would be important to support this anti-inflammatory activity of PZA through PARP1 in complementary ways, namely, by presenting histologic analysis and quantification of the lungs, as shown in the previous set of experiments. Similarly, in Fig 6 only bacterial burdens are shown, but given the links presented in the rest of the study, it would be important to address the immune response and lung pathology in this case.*

We thank the reviewer for these concerns, and acknowledge that PZA may affect PARP-dependent and -independent host immune responses (lines 252-256, 270-275, 291-293, and 400-415). We did not preserve lung tissue for histology in either of these studies, and hence complementary histological evaluation is unfortunately beyond the scope of what we can do. However, based on the reviewer's suggestion we added 6 panels of cytokine levels (new Supplementary Figure 10) in PZA-treated WT and PARP1^{-/-} mice infected with PZA-susceptible *M.tb H37Rv* (corresponding to the experiment in Figure 6). TNF α , IFN γ and IL-1 β were significantly reduced in WT but not PARP1^{-/-} mice, while IL-6 and IL-10 did not change significantly in either group and MCP-1 was reduced in both groups. The results were not overwhelming (PZA lowered cytokines in both WT and PARP1^{-/-} mice, which is not surprising considering it also reduced bacterial burdens in both groups of mice) but since the difference in several cytokines (TNF α , IFN γ and IL-1 β) only reached significance in WT but not PARP1^{-/-} mice the anti-inflammatory effects of PZA are likely at least in part mediated by its interaction with PARP1. These findings are in line with all of our other data and suggest that PARP1 inhibition is *one* mechanism that contributes to PZA-mediated anti-inflammatory effects, in addition to the anti-inflammatory consequences of reducing the bacterial burden.

In order to eliminate confounding changes in bacterial burden and solely focus on the host-directed activity of PZA, we focused on the effects of PZA in WT and PARP1^{-/-} mice infected with the PZA-resistant *pncA* mutant (Figure 5, new Supplementary Figure 9 [previously Supplementary Figure 5]). This study more clearly demonstrated that the host-directed anti-inflammatory effects of PZA are largely PARP1-dependent. Since PZA seems to have negligible bactericidal activity in TB patients (Xe et al, 2021, DOI: 10.1126/scitranslmed.abd7618), mouse infection studies with a PZA-resistant *M.tb* mutant may also more accurately portray PZA's activity in humans. We have addressed this in the text (lines 382-389).

Reviewer 2 (Remarks to the Author):

In this study, the authors suggest that inhibition of the inflammatory cellular regulator, PARP1, could support and enhance the in vivo effect of pyrazinamide (PZA) in M. tuberculosis (Mtb) infected mice by reducing inflammation. The work is divided into an in vitro part using recombinant PARP1 or monocytic cell lines, and an in vivo part involving Mtb infection of C3H mice and treatment with the first-line anti-TB antibiotics, PZA and rifampicin (RIF), in the presence or absence of the PARP1/2 inhibitor, talazoparib (Tp). The in vivo work also uses relevant Mtb mutant stains (PZA-resistant) and PARP1 deficient mice to investigate the single- and combined effects of Tp and PZA and/or RIF. The authors conclude that PZA, similar to

commercial PARP1 inhibitors, binds to the active site of PARP1. PZA further inhibits PARP1 activity in THP-1 cells and in *Mtb*-infected mouse lungs after 2-months *in vivo* administration. Treatment with *Tp* alone or in combination with RIF, reduces lung inflammation and pathology in *Mtb*-infected mice, even though bacterial loads in the lungs are relatively higher compared to other treatment groups. Luminex multiplex of lung homogenates shows that *Tp* treatment results in down-regulation of IL-1 β and IL-12, while a PARP1-dependent effect on a number of other cytokines and chemokines involved in inflammation/Th1 immunity is also proposed (non-significant observations).

Overall, the hypothesis that inhibition of a protein involved in inflammatory processes could be targeted for potential host-directed therapy in TB is interesting and highly relevant in a time of increasing antibiotic resistance. I also appreciate these difficult and time-consuming experiments with virulent *Mtb*, which are very important to obtain new knowledge of how immune responses in TB can be modulated. However, I'm not convinced by the data itself (please, see my specific comments below) and would like to see additional experiments to support the author's conclusions.

Potential limitations of the study and results should be included in the Discussion. Moreover, I consider the discussion around proinflammatory cytokines in TB to be a bit insufficient and too general. The findings should be balanced with an introduction and discussion around proinflammatory vs Th1 immunity and anti-inflammatory responses in TB, as inflammation is also a necessary requirement to prime both macrophages and specific T cell responses that could enhance intracellular eradication of *Mtb*.

We thank the reviewer for the suggestions. We have now included limitations of our study in the discussion (lines 390-399) and elaborated on TB immune responses in the introduction and discussion (lines 44-55, 331-359).

Specific comments:

1. I have several comment and questions with regards to the *in vitro* experiments in Figure 2, using monocytic cell lines and the PARP1 activator, MNNG. How come the authors didn't compare uninfected to *Mtb*-infected macrophages to confirm that PARP activity is upregulated in infected cells and that PZA, similar to *Tp* and NAM, could down-regulate PAR formation? This would be the logical experiment before studies of *in vivo* *Mtb*-infected mouse lungs.

We completely agree with the reviewer on the need to first evaluate PARP activity and inhibition *in vitro* in *M.tb*-infected cells. In fact we did perform such experiments, but unfortunately we found that the PAR signal in infected macrophages was below our limit of detection. This occurred despite numerous attempts to achieve a signal on the immunoblot by using various virulent or attenuated strains, including *M. bovis* BCG, *M.tb* H37Rv or H37Ra, and the rapid-growing *M. smegmatis*; MOI 1, 5 or 10; with or without IFN γ priming; and timepoints ranging from 30 min to 48 hours post-infection. Since we were unable to detect PARP activation in *M.tb*-infected cell lines, we could only assess the effects of PZA/PARP inhibition during infection *in vivo* and in cells stimulated with a PARP1 activator, cytokines or bacterial antigen (new Supplementary Figure 2). We have mentioned this in the text lines 390-393 under limitations of the study.

2. In addition, I believe these data would be significantly stronger if monocyte-derived macrophages were used instead of cell lines. There are standard protocols for this, and the authors have access to PBMCs from healthy donor blood, so this should be feasible.

We thank the reviewer for this comment. We agree that it would be interesting to evaluate PARP inhibition by PZA in primary cells but given that PAR levels are undetectable even in un-inhibited infected cells we would not expect interpretable results from such an experiment. Although we were unable to detect a PAR signal in *M.tb*-infected cells, we examined PARP1 activation elicited by cytokines (TNF α), bacterial antigen (LPS) and MNNG in different cell lines as well as **primary human monocyte-derived macrophages** (differentiated from PBMCs). This is shown in new Supplementary Figure 2 (previously only shown for MNNG). We found that primary cells responded well to all the stimuli tested (better induction than seen with THP1 or RAW264.7 cells, less background than J774.A1 cells), confirming that PARP1 activation in response to antigen or cytokine stimulation appears to be a conserved response that is retained in immortalized cell lines (in particular, in THP-1 and J774 cells). While we admit that cell lines often do not recapitulate the functionalities of primary cells, this does not seem to be the case for PARP1 responses. We therefore believe that differentiated THP-1 macrophage-like cells were an appropriate model for our *in vitro* characterization of PZA's effects on PARP1 activity. We indicate this in lines 131-136 in the text.

3. *Viewing the immunoblots of PAR in Fig. 2b, it is not clear to me where specific band(s) are located? The lanes look more like smears rather than showing specific band as for example in Fig. 3b. Is it possible to visualize the molecular weight control as well, for clarity? At least, the authors should discuss the difference in the bands obtained with cell lines in vitro and with lung homogenates in vivo.*

We thank the reviewer for this comment. Reviewer 1 also brought up this concern, and we have provided our response above (response to reviewer 1, comment 1), and in the text (lines 127-129, and 393-397). In short, our PAR antibody is specific for target-bound ADP-ribose polymers, which vary greatly in size, so the resulting blot should be a broad smear rather than a defined band. The primary target of PARP1 is PARP1 itself, so as the magnitude of activation (or detection) decreases the range of the smear becomes narrower, with the most pronounced focus around 116 kDa (representing PARylated PARP1). A weaker signal thus can have the appearance of a "clean band" on standard Western blots, however the ideal/preferred PAR blot resembles those depicted in Figure 2 (and Supplementary Figure 2).

4. *Figure 3b illustrates that there is obviously a great variability in PAR formation comparing Mtb-infected individuals ie. n=5 mice/group. Is that expected and something that has been shown in other studies/diseases?*

We thank the reviewer for this comment. We have analyzed this further and found that PAR levels 1 month post infection in fact correlated with bacterial burden, but not with body, lung or spleen weights. We have now indicated this in lines 149-152 in the text. We also included the regression plots in new Supplementary Figure 3 (panels c and d).

5. *In Figure 4b, why wasn't a group with Tp+PZA included? Perhaps I missed out on something, but it is quite obvious that this group should have been included in the CFU graphs similar to Tp+RIF. Even if the difference in CFU counts between RIF alone and Tp+RIF is not significant, it is quite clear that the latter group has relatively higher CFU counts. I'd appreciate an extended discussion on this phenomenon.*

We thank the reviewer for this concern. We used various approaches to tease the host-directed activities apart from PZA's bactericidal activity, including infection with a PZA-resistant strain of *M.tb* and the use of PARP1-deficient mice. In C3HeB/FeJ mice, we interrogated the effects of bacterial killing and PARP1 inhibition individually and in combination to determine how each

factors into the antimycobacterial activity of PZA. The combination of RIF+Tp is meant to mimic the dual activity of PZA (i.e., bactericidal activity + PARP1 inhibition), which we now more explicitly state in lines 189-192. Since either Tp or PZA were sufficient to reduce PARP1 activity to uninfected levels, we did not feel the need to include a Tp+PZA group.

We agree with the reviewer's observation regarding increased CFU observed with Tp (alone or with RIF). Further analyses (new Supplementary Figure 4 b, d-e) confirm that PARP1 inhibition without adequate antibiotics appears to antagonize bacterial clearance and have addressed this in the text (lines 196-198). These analyses showed that in the absence of effective antibiotics (vehicle, Tp), PAR levels were trending toward an inverse correlation with CFU (new Supplementary Figure 4 e), suggesting that PARP1 may contribute to TB containment during the chronic phase of infection (lines 331-336). Impaired containment may be due to a reduction in protective CD4 T-cell responses or macrophage activation (reduced CD4+ T cell frequency, new Supplementary Figure 6 d; reduced CXCR5 or iNOS expression, new Supplementary Figure 8 b). However, RIF + Tp-treated mice still had significantly reduced bacterial burdens compared to vehicle-treated mice, along with dramatically improved lung pathology compared to all other treatment groups (lines 208-210). We thus consider our study a proof-of-principle evaluation of adjunctive PARP inhibition in TB therapy and are optimistic that future studies can further optimize the Tp dosing strategy to minimize its negative impact on bacterial clearance while preserving its ability to accelerate the resolution of TB lung disease (discussed in lines 355-357). In addition, we modified our proposed model (Figure 6 e) to indicate PARP1's role in TB control and the potential risk of PARP1 inhibition without adequate antibiotic activity.

6. *Inflammation and lung involvement presented in Figure 4d-e, are two important criteria used to illustrate that inhibition of PARP1 activity results in reduced pulmonary pathology in TB disease. In the Materials and methods, it should be explained in more detail how these parameters were quantified. Here, I would recommend to use a validated scoring or grading system to authenticate the data.*

We thank the reviewer for pointing this out. We have provided additional details of the criteria for scoring lung inflammation and pathology in our Methods section (lines 564-570). These analyses were performed by a board-certified veterinary pathologist with expertise in the histopathology of TB lung disease in laboratory animals. Quantitative analysis using ROI selection is superior to the use of validated scoring systems, which are only semi-quantitative, inherently more subjective, and done when quantitative analysis cannot be performed.

7. *In addition, only H&E stains were used to study inflammation and lung involvement, while immunostainings for eg. neutrophils (elastase), CD4 and CD8 T cells, and myeloid cells (inflammatory and anti-inflammatory monocytes and macrophages), would significantly strengthen these data. The authors have access to formalin-fixed lung tissue so this should be feasible. An interesting option that is perhaps out of the scope of this manuscript, is also to use lung homogenates for flowcytometry, to obtain a systematic view on the immune cell subsets present in the lungs after respective treatment. These analyses would enable a more detailed assessment of immunopathology in Mtb-infected mice treated with PZA, Tp etc.*

We thank the reviewer for this helpful suggestion. In response, we have now done immunostaining for macrophages (F4/80), neutrophils (Gr-1), fibroblasts (vimentin), CD4+ T cells (CD4) and CD8+ T cells (CD8). Our new findings are presented in new Figure 4 f and new Supplementary Figures 6-7. We found that RIF+Tp or RIF+PZA treatment dramatically reduced neutrophil frequencies in the lung and granuloma (Supplementary Figure 6 d-e). Surprisingly,

neutrophil frequencies were unaffected by RIF alone, even though RIF-treated mice had lower bacterial burdens than RIF+Tp-treated mice, or by Tp alone, indicating that 1) neutrophil frequencies were not reflective of bacterial burden and 2) adjunctive PARP inhibition can reduce neutrophil infiltration only when bacterial replication is controlled by antibiotics. This is keeping with the emerging understanding of the role of PARPs in regulating innate immune responses, especially in the recruitment and function of neutrophils (Szabo et al., 1997, DOI: 10.1084/jem.186.7.1041; Zhu et al., 2021, DOI: 10.3389/fimmu.2021.712556). In contrast, lung infiltration by fibroblasts, macrophages and CD8+ T-cells decreased in response to either bacterial killing or PARP inhibition, and most potently to both.

In addition, PARP1 appears to be critical for CD4+ T-cell responses or expansion since Tp and RIF+Tp-treated mice had the overall lowest CD4 frequencies. Considering their protective role in TB, this observation offers a potential explanation for the impaired TB containment associated with PARP inhibition. Similarly, PZA or RIF treatment increased the expression of CXCR5 in lungs, a response that was attenuated by the addition of Tp (new Supplementary Figure 8 b). Since CD4+CXCR5+ T cells have been shown to promote protective immunity against tuberculosis (Slight et al., 2013, DOI: 10.1172/jci65728), we speculate that an overall reduction in protective CD4+ T-cell responses may be responsible for the increased bacterial burden in RIF+Tp-treated mice. These findings are discussed in the text (lines 215-226, 241-258, 336-339, and 344-348).

8. *Regarding Figure 5c-d, and as explained in the summary, I'd like to obtain a nuanced description of the role of ie. IL-1b (activation of neutrophils) compared to IL-12 (activation of DCs), TNFa (activation of macrophages) or IFN-g (activation of macrophages and T cells). As far as I understand it, there is a trend towards down-regulated levels also of IL-10 in WT and PARP1-/- mice (Supp. Fig. 5). Are any cytokines/chemokines or other mediators up-regulated by PARP1 inhibition? What is the definition of pathological (pro)inflammation in TB and when or how is it important to block inflammation? Assessment of other inflammatory mediators known to be important in Mtb-infected cells, such as iNOS/NO, ROS, or autophagy, would also add to an understanding of the function of PZA in PARP1-dependent regulation of TB immunity.*

We thank the reviewer for these highly relevant points. We have substantially extended our discussion of TB immune responses as suggested, in the introduction (lines 44-55, 76-79) and discussion (lines 336-359). We did not find any genes we looked at to be upregulated by PARP1 inhibition, but IFN γ , IL-1 β and RANTES were slightly higher in PZA-treated than in untreated PARP1^{-/-} mice, and IL-6 levels were slightly higher in PARP1^{-/-} than in WT mice but unaffected by PZA (Supplementary Figure 9). To get a better sense of the immunological consequences of PZA or PARP1 inhibition in TB therapy, we have added several additional figures evaluating relative immune cell frequencies (including macrophages, CD4+ T cells, CD8+ T cells, neutrophils and fibroblasts) by IHC (new Figure 4f; Supplementary Figure 6 c-e; Supplementary Figure 7) and transcriptional changes (including *Tnfa*, *iNOS*, *Mcp-1*, *Cxcr5*, *Ifnb*, *Ifit1* and *Ifit3*) by qPCR (new Supplementary Figure 8b). These analyses proved to be quite insightful and revealed that combined antibiotic activity with PARP1 inhibition potently reduced neutrophil frequencies in the lung and granulomas, and that fibroblast frequencies correlated with the dramatic reduction in immunopathology we previously observed. In addition, we found that adjunctive PARP1 inhibition, more than changes in bacterial burden, lowered the expression of *Tnfa*, *iNOS*, and type 1 IFN signaling (*Ifnb*, *Ifit1*, *Ifit3*). The positive association between type 1 IFNs and neutrophils with TB progression and immunopathology led us to propose type 1 IFN signaling as a primary target mediating the effects of PARP1 inhibition. We also added the cytokine analysis for the experiment in Figure 6 (infection of WT and PARP1^{-/-}

mice with PZA-susceptible *M.tb* H37Rv; new Supplementary Figure 10), but as expected the bactericidal activity of PZA limits the usefulness of these data.

9. *The illustration in Figure 6e is nice, but adjunctive PARP1 inhibition with ie. Tp+RIF, would then result in intermediate/relatively higher Mtb load but reduced lung damage, perhaps this should be included somehow.*

We thank the reviewer for this helpful comment. We now indicate that PARP1 activation promotes bacterial containment in our revised model (Figure 6 e) and state that PARP1 inhibition without adequate antibiotics may impair bacterial control in the figure legend.

10. *My final comment is that it that it would have been good to obtain viability data from the mice used in the respective groups, especially given the slightly higher bacterial loads in Tp+RIF compared to RIF alone. I realize this would require a separate experiment, but perhaps this could be done for the most interesting groups/drug combinations. Even if PARP1 inhibition can reduce immunopathology in TB, it is still not clear to me how PARP inhibitors could be used as adjunct treatment in MDR-TB patients, in the absence of PZA (as mentioned in the Discussion).*

We thank the reviewer for this suggestion. In our initial studies, we did find that a higher dose of Tp indeed impaired *M.tb* containment and resulted in the death of some infected mice. We subsequently reduced the Tp concentration to 0.5 mg/kg, which did not induce any notable toxicity or death even in infected mice, and used that concentration for all experiment presented in this manuscript. We would like to point out that even though the bacterial burden in RIF+Tp was slightly higher than with RIF alone, it was still 2 logs lower than vehicle-treated mice (lines 196-199 and 208-210), indicating that bacterial replication was still well-controlled and that the infected mice would continue to survive for a long time (likely until RIF inevitably becomes ineffective due to resistance). Without combination therapy, drug resistance will eventually arise in any mouse treated with PZA monotherapy or RIF monotherapy (+Tp), making it tricky to evaluate survival effects with confounding drug resistance.

We view this manuscript as an exciting insight into the host-directed mechanism of PZA and a proof of principle that adjunctive PARP1 inhibition has the potential of reducing lung inflammation in the context of TB. More work will need to be done on the effectiveness of PARP inhibitors especially in MDR/XDR TB patients in whom PZA resistance is common. We address this in the limitations of the study (lines 355-357 and 397-399).

11. *Statistical comment: Please, note that data not passing a normality test should be presented (median +/- IQR) and analysed using non-parametric methods.*

We thank the reviewer for this observation, and have now confirmed that most of our data falls within a normal distribution by D'Agostino & Pearson normality test. Any data not following a normal distribution (IL-6 lung levels in Supplementary Figure 9; all cytokines in Supplementary Figure 10) are now analyzed by nonparametric Kruskal-Wallis test with uncorrected Dunn's multiple comparisons test, and graphed showing median +/- IQR as suggested. We have also updated our description of statistical analyses in the methods section (lines 600-603). Note that some groups had sample sizes that were too small to determine normality.

Reviewer #3

Krug et al. identify PARP1 as a host factor that could be targeted by the anti-TB compound pyrazinamide (PZA), at least partly explaining the effects of PZA use. The authors indicate that there is literature supporting PZA acting against a host target to influence the inflammation response during treatment, but that the target has not been identified. The study first investigates whether PZA will bind to the PARP1 active site, and they test whether PZA lowers poly(ADP-ribose) production in cells. Using mouse models infected with TB, the study tests the influence of PZA (alone or in combination with other compounds) on poly(ADP-ribose) production, bacterial burden in lungs, lung histopathology, and immune response. The overall story is compelling; however, the presentation of the data and limitations could be improved.

Here are some specific comments:

- 1. Figure 1c. What do the fluorescence v. temperature plots look like (rather than the derivative of the fluorescence signal)? The derivative plots typically will have a stronger single peak. It seems these experiments were performed with the entire PARP1 molecule (not just the catalytic domain), and this could perhaps explain the appearance of the data, since full-length PARP1 will have several domains unfolding and not necessarily at the same temperature. The errors associated with the T_m measurements should be shown, and the plot of the raw data would be helpful. It would also be helpful to perform an *in vitro* inhibition assay with these compounds, rather than moving directly to cells.*

We thank the reviewer for this keen observation. We now present the fluorescence vs. temperature plots of PARP1 with NAM or PZA in new Supplementary Figure 1. We indeed observed multiple peaks representing the unfolding of different PARP1 domains. However, only the 46°C peak was affected by NAM or PZA binding, similar to what has been previously described for other small molecule inhibitors such as BAD that bind the PARP1 active site (Langelier et al., 2018, DOI: 10.1038/s41467-018-03234-8). This is now indicated in lines 119-122. We agree with the reviewer that a direct *in vitro* inhibition assay could have been helpful and provided additional evidence that PZA inhibits the catalytic activity of PARP1 but we felt that our data at the time warranted moving directly into more biologically relevant settings (macrophages, TB-infected mice), where we were able to demonstrate that PZA indeed inhibits PAR formation induced by TB infection or MNNG/PARP1 activation.

- 2. Page 6, the description of Supplementary Figure 2 requires some nuance, since the PAR levels were quite variable in infected lung cells. It seems that there is no statistical difference, but the increase is referred to as robust.*

We thank the reviewer for this observation. While we agree that there is considerable variability in PAR levels, PAR levels are a minimum of 2- to 5-fold higher in the lungs of *any* infected mouse 1 month post-infection than the average PAR intensity in uninfected lungs. Despite the distribution, and the infrequent mouse with inexplicably high PAR levels at baseline, this difference was statistically still highly unlikely to occur by random chance even though the *p* value (0.0564) did not quite make the “significance” cutoff. Especially given the variability, we do consider a 2-5-fold increase “robust” but have now removed this modifier from the text (lines 147-149).

Importantly, we have now, and as suggested by reviewer 1, further analyzed the relationship between PAR levels and bacterial burden, body weight, lung weight or spleen weight to gain insight into the observed variability (new Supplementary Figure 3 c-d). We were excited to find

that PAR levels at this time point correlated strongly with the corresponding bacterial burden ($r^2 = 0.9161$, $p = 0.0106$) but not with any of the other variables, indicating that the variability in lung PAR levels is best explained by differences in bacterial burden 1 month post-infection. We have discussed these findings in the text (lines 149-152 and 331-333).

3. *“By structural alignment, we found that PZA is predicted to bind the PARP1 ART fold in the same manner as BAD, and that it even forms one additional bond (Phe897) (Figure 1 b, right)”*

I would tweak this sentence in the following way to be more accurate:

*“... and that it is predicted to form an additional bond (Phe897) based on our modeling...”
The model seems reasonable, but even small changes can lead to massive rearrangements in how molecules bind to active sites, so it is important to stress that a model is being presented.*

We thank the reviewer for this kind suggestion. We have made the change in the text as suggested (lines 112-113).

4. *Figure 3, there needs to be some discussion of the variability observed within treatment groups. What can explain the Tp-treated mouse with substantial PAR levels, for example? The results do not seem to be as clear cut as the presentation in the text. For the statistical analysis, it would be useful to list all of the P values for the comparisons made.*

We thank the reviewer for this observation. In response to this concern, and as suggested by reviewer 1, we have now analyzed the relationship between PAR levels and bacterial burden by treatment group (new Supplementary Figure 4 b) and added multiple panels of regression plots (Supplementary Figure 4 c-e). While we cannot explain *why* PARP1 activity was incompletely suppressed by Tp in some mice, or why one vehicle-treated mouse had much lower PAR levels than the other control mice, we found that *all* Tp-treated and all-but-the-one vehicle-treated mice formed well-defined clusters in terms of PAR and CFU when grouped by treatment, with PAR levels in *all* Tp-treated mice falling outside of (below) the PAR levels in vehicle-treated mice (Supplementary Figure 4 b). These findings indicate that PARP1 was nonetheless inhibited to an activity level less than the level in untreated mice in all Tp-treated mice. We further found that PAR levels in mice not receiving antibiotics (vehicle and Tp) were trending toward a weak inverse correlation with bacterial burden (Supplementary Figure 4 e). Since PARP1 activity at 1 month post-infection (at the peak of bacterial proliferation) was directly proportional to bacterial burden, PARP1 may be activated by *M.tb* directly or indirectly by a consequence of replication. If, as our data suggest, PARP1 activity contributes to bacterial containment in the chronic phase of infection, Tp may antagonize this and allow more bacterial replication, which would again stimulate PARP1 activity, offering a potential explanation for the persisting PARP1 activity in some Tp-treated mice.

These analyses also provided additional insight into the activities of PZA and RIF. While the bactericidal activity of RIF was independent of PAR levels, PZA treatment was unique in generating 4 distinct outcomes: PZA reduced both PAR and CFU in 4/8 (50%), PAR only in 2/8 (25%) and CFU only in 1/8 (12.5%) mice, while affecting neither in 1/8 (12.5%) mice. In fact, 80% (4/5) of the mice in which PZA reduced bacterial burdens also had potentially reduced PAR levels, indicating that PZA efficacy most often coincides with PARP1 inhibition. These findings suggest that PARP1 inhibition is not required for but may potentiate the bactericidal efficacy of PZA. We have elaborated on these findings in the text (lines 160 – 179; lines 377-381).

While we chose to show symbols instead of p values in some figure panels that would otherwise appear too crowded, we will list all *p* values in the source data file.

5. *Page 6, "and no discernable inhibitory effects at eight times the concentration used to treat mice (Supplementary Table 1)" Is this data actually shown? I could not find it.*

These data are shown in Supplementary Table 1, included in the original submission and now in the revised manuscript as well.

6. *Figure 5, why not just show the P values across all samples?*

We thank the reviewer for this concern. Due to space constraints, we are unable to include *p* values in the figure but we will include all statistical analyses in the source data file.

7. *Figure 5 data. These mouse experiments were done using TB with resistance to PZA in order to focus on the host effects of PZA. If the only host effect was to target PARP1, it seems that PARP1 knockout should mimic PZA treatment of WT mice. However, the PARP1 knockout mice appear to be quite different from PZA-treated WT mice. There might be a good explanation for this, but I found this to be confusing and counter intuitive.*

We thank the reviewer for this valid concern. Lack of the PARP1 gene likely has many effects on cell-mediated and humoral immune responses in mice and also on the development of the immune system that may impact TB responses differently than simply inhibiting PARP1 only for the course of 8 weeks after the infection is already established. Our new analyses indicate that PARP1 activation may promote bacterial proliferation in the acute phase of the infection while activation in the chronic phase may contribute to bacterial control but also to lung inflammation (discussed in lines 331-335). Not having PARP1 during all stages of infection thus may impact infection kinetics and host responses differently than inhibiting PARP1 during only the chronic phase. The fact that lung bacterial burden is slightly (non-significantly) lower in PARP1^{-/-} than in WT mice before that start of treatment and slightly (non-significantly) higher at the end of treatment (in untreated mice) somewhat corroborates this explanation.

Despite these known developmental immune deficiencies (see for example Rosado, 2013, DOI: 10.1111/imm.12099), PARP1^{-/-} mice were nonetheless capable of containing the infection and mounting immune responses that were largely comparable to WT mice, indicating that other processes can compensate for the absence of PARP1 to restore effective host responses. While we don't know what these compensatory mechanisms are, our observation that cytokine levels in PARP1^{-/-} mice were largely unaffected by PZA implies these other factors are not targets of PZA and unlikely to contribute to PZA's mechanism of action.

Alternatively, it is still possible that PZA also modulates PARP1-*independent* host mechanisms (mentioned in lines 270-273). In addition, Tp inhibits not only PARP1 but also other members of the PARP family, most notably PARP2 (which is structurally quite similar to PARP1 and may compensate for PARP1 absence in KO mice). If PZA has similar broad-spectrum PARP-inhibitory activity, differences could thus also be explained by the lack of PARP1 vs. the lack of multiple PARPs, some of which may even play PARP1-compensatory roles. We now discuss this in lines 400-411.

8. *Along the same lines, can it be assumed that the PARP1-directed effects of PZA are captured in Figure 6d, by comparing PZA-treated WT to PZA-treated PARP1^{-/-} ? I was left*

without a strong feeling for how much the PARP1-directed effects of PZA contribute. Perhaps this can be elaborated on.

This is an excellent point. Hypothetically, if we were to assume there were 1) no timing differences (i.e., missing PARP1 throughout as opposed to simply inhibiting it during chronic infection), 2) no PARP1-compensatory mechanisms and 3) no PARP1-independent host-directed activity of PZA, we would expect PZA to work as well (or better, assuming that there could still be *some* low-level PARP1 activity in PZA-treated WT mice) in PARP1 knockout mice than in WT mice. However, as explained in our response to the preceding comment, our data indicate that some unknown, *PZA-unresponsive* factors compensate for the lack of PARP1 in TB-infected PARP1^{-/-} mice. The data in Figure 6 suggests that without its ability to modulate PARP1-dependent responses the bactericidal efficacy of PZA is diminished.

We have now also analyzed cytokine responses from this study (new Supplementary Figure 10), which demonstrate that bacterial killing alone is sufficient to dampen proinflammatory immune responses but that this effect was attenuated in PARP1^{-/-} mice (discussed in lines 285-289). Whether this a direct consequence of absent PARP1 modulation or an indirect one due to the higher bacterial burden is unclear, but the critical experiments with the PZA-resistant *pncA* mutant (Figure 5, Supplementary Figure 9) definitively demonstrate that in the absence of bactericidal activity, PZA's host-directed activity is largely PARP1-dependent. Since PZA has negligible bactericidal but potent anti-inflammatory activity in human TB patients (Xie et al, 2021, DOI: 10.1126/scitranslmed.abd7618), our data collectively suggest that PARP1 inhibition may be a key component of PZA's mechanism of action underlying its unique treatment-shortening ability in TB therapy. As requested by the reviewer, we now elaborate on this on lines 400 to 415.

9. *What is the basis for the term "sterilizing ability"? Does this refer to antibacterial activity? Worth a short sentence for those not familiar with this term.*

We thank the reviewer for pointing this out, and have now briefly defined the term sterilizing activity of antimycobacterial drugs in the introduction (lines 60-63).

REVIEWERS' COMMENTS

Reviewer #1 (Remarks to the Author):

The authors did a good job in answering the raised concerns, by adding a substantial amount of new data and improving their discussion of the results. I have only a couple of minor comments:

1) in lines 151-152, the authors suggest that "PARP1 activity might promote M.tb proliferation or persistence in the acute phase of infection" based on the finding that "PAR formation in response to M.tb infection correlated strongly with bacterial burden". I guess the counter argument may also be a possibility: that the increase in bacterial burden may enhance PAR formation. This is of course very difficult to distinguish, but raising the two possibilities may be appropriate.

2) in line 237, the statement "...while TNF α inhibition was only observed in the presence of antibiotics (Supplementary Figure 8 a)." is not correct, because TNF secretion is also inhibited in PZA alone.

3) in lines 353-354, the claim "Our findings implicate Type I IFN signaling as a primary target of adjunctive PARP1 inhibition" may be an overstatement. The authors find several alterations to adjunctive PARP1 inhibition, one of which is type I IFN. Despite being a very compelling candidate, I suggest to remove the word primary, as the data do not allow such a claim.

Reviewer #3 (Remarks to the Author):

The authors have addressed the major concerns that I raised during the initial review.

Reviewer #4 (Remarks to the Author):

In my opinion, the authors have adequately addressed the concerns raised by original Reviewer #2 by providing additional supporting experimental data, and substantial rewriting and additions to the introduction, results and discussion.

Importantly, this work provides a novel mechanistic answer to a long-standing and contentious mystery of the mechanism of PZA. In my opinion, this is a noteworthy finding by itself and should not be impacted by whether targeting PARP1 is clinically viable. That question is much more complex given the acute and chronic phase of infection. The concerns of the double-edged activity/function of PARP1 in TB pathogenesis (also raised by other reviewers) is valid, but the authors have done an admirable job in teasing out parts of the mechanism and present the results and conclusions in an acceptable manner without overstating the host-directed contribution and clinical application. The addition of the limitations of the study are nicely integrated and adequately discusses all the key weaknesses of the study.

Re: **NCOMMS-22-0952A** – Inhibition of host PARP1 contributes to the anti-inflammatory and antitubercular activity of pyrazinamide

We would like to sincerely thank the reviewers for their time and constructive feedback throughout this review process. Thanks to their insightful suggestions, we feel that our final manuscript has greatly improved and now provides more mechanistic insight into PZA's host-directed activity. We have made minor changes to the final document to address the concerns raised by Reviewer 1 as stated below.

Reviewer #1 (Remarks to the Author):

The authors did a good job in answering the raised concerns, by adding a substantial amount of new data and improving their discussion of the results. I have only a couple of minor comments:

1) in lines 151-152, the authors suggest that "PARP1 activity might promote *M.tb* proliferation or persistence in the acute phase of infection" based on the finding that "PAR formation in response to *M.tb* infection correlated strongly with bacterial burden". I guess the counter argument may also be a possibility: that the increase in bacterial burden may enhance PAR formation. This is of course very difficult to distinguish, but raising the two possibilities may be appropriate.

We thank the reviewer for this comment. We have changed the statement in line 151-152 to "This increase in PAR formation in response to *M.tb* infection correlated strongly with bacterial burden but not with lung, spleen or body weights of mice (Supplementary Figure 3 c-d), suggesting that PARP1 activity is enhanced by increasing bacterial burdens and might promote *M.tb* proliferation or persistence in the acute phase of infection."

2) in line 237, the statement "...while TNF α inhibition was only observed in the presence of antibiotics (Supplementary Figure 8 a)." is not correct, because TNF secretion is also inhibited in PZA alone.

We thank the reviewer for this comment, and have now clarified the statement to specifically include both RIF and PZA. We still consider PZA an antibiotic despite its weak bactericidal activity but agree that our previous statement was potentially misleading.

3) in lines 353-354, the claim "Our findings implicate Type I IFN signaling as a primary target of adjunctive PARP1 inhibition" may be an overstatement. The authors find several alterations to adjunctive PARP1 inhibition, one of which is type I IFN. Despite being a very compelling candidate, I suggest to remove the word primary, as the data do not allow such a claim.

We thank the reviewer for this comment, and have now removed the word "primary" as suggested.

Reviewer #3 (Remarks to the Author):

The authors have addressed the major concerns that I raised during the initial review.

Thank you!

Reviewer #4 (Remarks to the Author):

In my opinion, the authors have adequately addressed the concerns raised by original Reviewer #2 by providing additional supporting experimental data, and substantial rewriting and additions to the introduction, results and discussion.

Importantly, this work provides a novel mechanistic answer to a long-standing and contentious mystery of the mechanism of PZA. In my opinion, this is a noteworthy finding by itself and should not be impacted by whether targeting PARP1 is clinically viable. That question is much more complex given the acute and chronic phase of infection. The concerns of the double-edged activity/function of PARP1 in TB pathogenesis (also raised by other reviewers) is valid, but the authors have done an admirable job in teasing out parts of the mechanism and present the results and conclusions in an acceptable manner without overstating the host-directed contribution and clinical application. The addition of the limitations of the study are nicely integrated and adequately discusses all the key weaknesses of the study.

Thank you!